# Threat Utility of the Seaport Risk Factors: Use of Rough Set-Based Genetic Algorithm

**Muhammad Reza Do Bagus** [ID] **and Shinya Hanaoka \***[ID]

Department of Transdisciplinary Science and Engineering, Tokyo Institute of Technology, Tokyo 152-8550, Japan
\*  Correspondence: hanaoka@ide.titech.ac.jp; Tel.: +81-3-5734-3468

**Abstract:** The threat due to risk factors disrupts supply chain continuity. To ensure supply chain continuity, it is important to understand the interdependency between seaport risk factors and the threat of supply chain disruption, from an economic and risk management perspective. This study understands the threat utility of port-centric supply chain risk disruption (PSCRD). It proposes a rough set-based genetic algorithm model and adopts the hybrid-conjoint analysis concept to generate the threat utility function. It is the sum of the level of disruption by the conditional seaport risk factors influencing the satisfaction of seaport-fulcrum supply chain continuity. It selects Indonesia to illustrate PSCRD empirically. Based on 153 samples of experts' evaluation, the rough set model highlights 24 conditional seaport risks as central tendency risk factors and classifies them into ten-dimensional threat factors. The results show that the seaport service process threat is the primary source of PSCRD in Indonesia; it reduces utility satisfaction to 32.2%, in the 100% utility estimation. This is followed by the relationship and planning process threats with 28% and 26.6% utilities, respectively. This study presents a framework to analyze PSCRD in relation to utility satisfaction and demonstrates the need for an integrated plan to enhance SSC resilience.

**Keywords:** seaport risk analysis; supply chain threat utility; rough set theory; hybrid-conjoint analysis

## 1. Introduction

The last decade has highlighted the importance of understanding the potential threats of seaport operations to supply chain continuity. These threats have implications in several dimensions and give rise to other threats, which weaken supply chain resilience. In this context, it must be noted that seaports are vulnerable to severe hazards that lead to a wide range of risks, including operational, environmental, security, technological, and organizational risks [1,2]. Deficient seaport operations result in supply chain disruption and negatively impact organizational profitability and reliability. It is critical to identify the causes of disruption originating from seaport operations to reduce the impact of disruptions, given the increasing integration of seaports into supply chains [3]. The several seaport risk factors and their resulting impact on supply chain continuity have increased the importance given to seaport operations.

The traditional seaport business is very labor-intensive and shares an interdependent relationship with other supply chain entities. The process, by which this relationship is established, poses a dimensional threat to the seaport supply chain, and hence plays a crucial role in seaport disruption events. This relationship process is associated with manpower in terms of soft skills and is affected by governmental policies. For instance, inefficient maritime security inspections at sea increase the shippers' exposure to liabilities under the contract of carriage and lead to operational delays, which, in turn, lower the promptness of goods [4,5]. Similarly, miscommunication prevents the effective execution of instructions [6], and labor shortages delay cargo handling [5]. Moreover, a conflict of interest between parties hinders decision-making processes and disintegrates the supply chains [7]. Conventionally, a risk matrix is adopted to manage these supply chain risks.

This risk matrix is classified into independent categories such as physical, financial, information, relationship, and innovation threats [8–10]. However, Ref. [11] argued that risk assessment/evaluation and risk treatment following the conventional risk identification techniques fail to account for complex dynamics across risks and risk sources, and hence yield sub-optimal solutions. Thus, we define the port-centric supply chain risk disruption (PSCRD) as the sum of the disruption level by the conditional seaport risk factor that influences the satisfaction of port-centric supply chain continuity.

This study comprehensively considers the multiple criteria decision-making (MCDM) with the utility theory as a risk management approach to understand the complex dynamics under risk criteria. The MCDM with utility function is useful to explain the optimal solution or rank a set of alternatives derived from predetermined risk profiles based on their performance on the set criterion [12]. To solve MCDM problems, it is essential to assess and aggregate the performance of each option on each criterion, which may be described as ordinal or cardinal information [13].

The functional requirement as a minimal constraint is an element that should be included in practical MCDM problems [12]. The MCDM problems contain both quantitative (e.g., price) and qualitative (e.g., risk) criteria. While the values of quantitative criteria can be scored as a crisp or interval number, it is difficult to express the values of qualitative criteria numerically. However, the latter can be represented directly in linguistic terms [10]. The direct aggregation of experts' opinions into collective opinions can provide a distorted picture of the diverse perceptions of experts. Hence, the trade-off achieved by integrating the subordinate ranking sets derived from each expert can be utilized to avoid the distortion of individualized evaluations in the aggregation of a utility function. In this regard, it must be noted that a large number of conditional seaport risk features make aggregation more uncertain or unsolvable.

In the computational process, this study effectively retains decision-makers' original opinions on seaport risk criteria and the combination of experts' evaluations. Specifically, it uses the rough set-based genetic algorithm (RSGA) and deploys the threat utility function to understand the utilization of PSCRD. Therefore, the identification of seaport risk involves understanding the supply chain threats due to conditional risk events. Correctly identifying those risks contributes to the logistics industry and shipping industry by increasing seaport resilience and ensuring business sustainability.

## 2. Literature Review

As an intersection between the worldwide mobility chain of goods and people, seaports have become critical to effectively and efficiently evaluate, as well as manage, PSCRD, protect the people and the environment, and maintain quality and performance. Recently, container shipping operations have acted as the backbone of global supply chain and have thus created a hotbed of multiple operational risks that have affected other supply chain entities [14]. For example, Ref. [15] found that the low punctuality of delivery goods due to inefficient maritime security inspections increased exposure to liabilities under their contract of carriage. As an effect, stoppages for maritime security checks at sea generate delays, which raise shipping expenses such as reschedule services; including pilotage, class inspections, and planned maintenance.

Another threat dimension is related to the organizing or planning processes. They play a significant role in the port-centric supply chain operations. However, these processes are susceptible to disruptions caused by several seaport risks, and hence capable of weakening the seaport supply chain's resilience mechanism. For example, Ref. [16] highlighted the problem of storage planning, specifically the inventory routing problem, in the fertilizer product supply chain. The inventory routing issue induces demurrage in the loading port, thereby declining the ship's utility and increasing its operational cost. A comprehensive planning process can help port-centric supply chain organizations to pursue the established direction with less uncertainty [17].

The seaport service process also represents a key threat dimension. The most commonly discussed PSCRD are seaport equipment breakdown [18,19] and inadequate equipment handling at the seaports [20]. Both these factors reduce the productivity and efficiency of seaport operations and expose the cargo to possible hazard and loss [21]. Eventually, these hazards might induce seaport accidents and reduce the ports' cost efficiency, thereby significantly influencing the total transport cost borne by the supply chain parties [22]. In terms of port-centric supply chain operations, Ref. [17] stated that the seaport service process risk is related to port services and distribution facilities. These include the rationality of port berth allocation; the operational efficiency of production (e.g., ship, crane, and container); and the efficiency and responsiveness in handling, storage, transfer, and distribution. These are inextricably tied to a yard, dock, and warehouse space design [23].

The distribution threats make up for another threat dimension. The transportation of dangerous goods (chemical spills) is considered a distribution threat [24]. Ref. [1] also explored some of the potential hazards associated with these distributions. They stated, for example, that handling hazardous goods or petroleum products might result in cargo spillages. Similarly, Refs. [25,26] mentioned specific hazardous events related to various sources. They include collisions due to the breakdown of cranes and trailers, potential leakage of substances and ignition resulting from the distribution of hazardous products, and contaminated premises. These hazardous events and delays and inefficiencies resulting from a poor infrastructure contribute to 10% of the cost of imported goods, as per data 2010 from the World Bank [27].

In the Indonesian context, an imbalance of cargo distribution, such as the availability of infrastructure, shipping patterns, supply and demand of maritime transport including port connectivity, between western part (developed economic region) and eastern part (developing economic region) make a challenge in the PSCRD. Ref. [15] addressed the issue of high logistics costs and price disparity between both regions. Moreover, this shipping cost harm the Gross Regional Domestic Product per capita in some part of developing economic due to the imparity above.

Concerning natural disasters, Ref. [24] analyzed the port of Boston and found that Category 3 and 4 hurricanes along with snowstorms can lead to severe economic and social consequences. Ref. [28], in particular, examined the impact related to seismic events and reported that the Kobe port in Japan suffered direct physical damage as a result of an earthquake. The damages amounted to more than USD 9 billion within the first nine months of the event that took place in 1995. These studies show that seaports are particularly exposed to extreme threats that may lead to various operational, environmental, natural, security, technical, and organizational risks [1,2].

The literature shows that seaports play a key role in supply chain continuity, given the increasing integration of seaports into supply chains. However, there is no explicit risk model to explain the interdependency between conditional seaport risk factors and potential threat of the supply chain, or to what extent the satisfaction level of seaport operation is due to the causal connection. Hence, a disruptive event originating at the seaports can hurt other interdependent businesses and alter the range as well as complexities of seaport service operations with other supply chain entities. An understanding of the seaport risk factors having a relationship with supply chain threats is necessary to elucidate on the characteristics of supply chain concerns, particularly the presence of numerous ports' dangers. Given this, Table 1 shows the issues potentially impeding seaport functioning and undermining supply chain continuity. By evaluating experts' responses objectively, this study proposes a risk model to investigate the association between the many conditional seaport risk factors and their threats to the supply chain continuity. Therefore, this study contributes to explain the causality connection among the many conditional seaport risk factors toward the potential threat of disrupted supply chain activities with a proposed framework to generate utility function. The utility function is then useful to explain the satisfaction level of the relative importance of the conditional seaport risk factors.

**Table 1.** Supply chain disruption threats under seaport risk events.

| No. | Seaport Risk Events | Threats to Supply Chain Disruption |
|---|---|---|
| 1. | The interdependency of disrupted events [14,15] | The creation of a domino effect that spread conditional seaport risk and increased the uncertainty of seaport operations resilience. |
| 2. | Organizational planning process [16,17] | Problem of storage planning, particularly in the fertilizer product supply chain<br>Decline in the ship's utility and increase in its operational cost owing to demurrage in the loading port |
| 3. | Seaport infrastructure breakdown [18,19,21,22] | Decline in the productivity and efficiency of seaport operations<br>Exposure of the cargo to possible hazard and loss<br>Decline in ports' cost efficiency and a consequential increase in the total transport cost borne by the supply chain parties |
| 4. | Operational inefficiency in terminal [17,21,23] | Impact on the planning of existing infrastructure and resources.<br>Impact on the availability of equipment and workforce |
| 5. | Inadequate cargo handling equipment for hazardous goods [1,24–27] | Cargo spillages caused by handling hazardous goods or petroleum products<br>Collisions due to the breakdown of cranes or rail-mounted cranes and trailers<br>Increase in the cost of imported goods |
| 6. | Natural disaster effect on the seaport operation [1,2,24,28] | Severe economic losses and social consequences<br>Exposure to operational, environmental, security, technical, and organizational risks |

Source: Made by the authors.

## 3. Framework of Threat Utility-Based Rough Set Design

### 3.1. Basis of Rough Set

The rough set theory (RST) approach is used to solve the qualitative MCDM problems by incorporating different linguistic representation models. The RST provides valuable tools for understanding data, quantifying and handling uncertainty, knowledge discovery, and handling vagueness in risk datasets. The rough set is combined with other methodologies in medical research [29]. It integrates a rough set-based genetic algorithm with a neural network to diagnose disease from clinical datasets. We use heuristic information, such as the genetic algorithm, to determine the central tendency of the risk factors and obtain a relatively minimal reduction among the seaport risk criteria from predetermining alternatives.

RST is a mathematical approach used for understanding and manipulating imperfect knowledge in the risk dataset without losing classification capabilities [30]. Given a decision table as a quadruple (4-tuples) $IS = \{U, A, V_a, f\}$, where $U = \{x_1, x_2, \ldots, x_q\}$ is a finite set of objects (universe); $A = \{a_1, a_2, \ldots, a_j\}$ is a finite set of features, consisting of conditional attributes and decision attributes denoted $A = C \cup D$, where $D = \{d_1, d_2, \ldots, d_j\}$, $A \in C$, and $A \notin D$; $V_a$ is the value set of attribute $a$, where $V = 1, 2, \ldots, 5$ indicates the highest to the lowest evaluation; $V = \cup_{a \in A} V_a$; and $f: U \times A \rightarrow V$ is a total function such that $f(x,a) \in V_a$ for each $a \in A$, and $x \in U$ is called the information function.

The indiscernibility relation ($I_B$), which is an equivalence relation, aims to reduce the geometric increase in the possible risk alternative and determine the elementary sets, connection, and functional form. For instance, some objects in $U$ (e.g., $x_1$ and $x_2$) can hardly be distinguished in an available set of attributes. Hence, let be $B$ in $A$. This definition is called an indiscernibility relation $I_B$ for every $b \subset B$. Any set of all indiscernible objects is called an elementary set as it represents the smallest discernible groups of decision-makers, and such a set forms a basic granule of knowledge about the PSCRD.

Moreover, the rule induction technique in the proposed model hinges on a pair of crisp sets known as the positive region ($\underline{B}(X) = \cup_{x \in U, I_B(x) \subseteq X} I_B(x)$ and negative region $\overline{B}(X) = \cup_{x \in X} I_B(x)$. The elements $\underline{B}(X)$ are all and only those objects $x \in U$ certainly

belonging to the equivalence classes generated by the indiscernibility relation $I_B$ contained in $X$. The elements of $\overline{B}(X)$ are all and only those objects $x \in U$ belonging to the equivalence classes generated by the indiscernibility relation $I_B$, containing at least one object $x$ belonging to $X$.

Formally, the dependence among the risk attributes may be defined as follows. If $D$ and $C$ are subsets of $A$, $D$ will depend on $C$ in degree $K$ ($0 \le k \le 1$), denoted by $C \Rightarrow _k D$, and if $k = \gamma(C, D)$. If $K = 1$, $D$ will depend entirely on $C$; if $K < 1$, $D$ will depend partially (in $K$) on $C$. These concepts of dependency are discussed in relation to the seaport risk factors in the datasets. In this case, $k = \gamma(C, D)$ is mathematically defined as $\gamma(C, D) = \frac{|POS_C(D)|}{|U|}$. Therefore, this concludes the basic concept of RSGA and indicates the removal of the superfluous attribute or remaining dependent attributes.

*3.2. RSGA for Features Selection*

The RST can be effective in solving the complexity and uncertainty of the port-centric supply chain risk, which removes the superfluous attribute to make the remaining attributes dependent. However, support is needed when dealing with seaport risks and nondeterministic polynomial-hard (NP-hard) problems in the combinatorial optimization of the dataset. Hence, a rough set-based genetic algorithm is designed to obtain the reduction attribute sets. These sets have a high degree of dependency and perform minimal data processing to solve the combination problem between the seaport risk and decision-maker evaluation of the PSCRD dataset.

Here, we consider a positive region in the RSGA scheme, in which the conditional seaport risk attribute set is $C = \{a_1, a_2, \ldots, a_j\} \forall B \subset C$. The dependency of the decision attribute on $B$ is defined as $\gamma(B, D) = \frac{|POS_B(D)|}{|U|}$.

To maintain the convergence speed and achieve the global optimum, while preserving the knowledge in the dataset, we propose a preliminary step to determine the pre-classification model as a significance attribute. In this model, the conditional attribute subset is $I_B \subset C \forall a_i \in I_B$, and the significance of $I_B$ is defined as $S_{a_n}(B) = \frac{|POS_C(D)| - |POS_{(B-a_n)}(D)|}{|U|}$.

For $a \in C$, the $S_B(D) = \gamma_C(D) - \gamma_{B-a_n}(D)$. If $k = 1$, $D$ will depend entirely on $C$, $S_B(D)$ will become larger, and $\gamma_{B-a_n}(D)$ will decrease, as mathematically defined by the positive region. Furthermore, the smaller the $\left|POS_{(B-a_n)}(D)\right|$, the greater the dependency will be of the decision attribute $D$ on attribute $a$ and the larger will be $\gamma(C, D)$, as mathematically shown in $S_C(D) = \frac{\left|POS_{(C-a_n)}(D)\right|}{|U|}$, where $|U|$ is a fixed value. Owing to the addition of attributes in the order of significance, a reduction may be obtained so that $Max\,(I_B\,(D)) \forall a_n \in \beta$. The set of *D-indispensable* attributes in $A$ is called *the D-core* of $A$. However, the term *D-reducts* is used to refer to the minimal subsets of conditional attributes that discern all equivalence classes of the relation $Ind(D)$, discernible by the entire set of attributes. In other words, if each conditional attribute in the decision table is independent of D, then the conditional attribute set $C$ will be independent of $D$; otherwise, $C$ will depend on $D$.

The higher the reduction in $B$, the higher the average of the significance attribute. This principle is reflected in the weight indicator $\overline{S}(B) = \frac{\sum_{i=1}^{|B|} S_{a_n}(B)}{|B|}$. For instance, let the conditional attribute set be $C$ and $B$ be a subset of $C$. As shown in the significance attribute, for $a_i$ element $B$, each attribute in $B$ is independent of $D$ and $\overline{S}(B)$ becomes larger. Otherwise, $\gamma_{B-a_n}(D)$ becomes smaller than that in the significance attribute. Thus, there is a higher likelihood of a reduction in $B$. Specifically, when $POS_B(D) = POS_C(D)$, $B$ is the reduction in $C$, according to the definition in the positive and negative regions.

The RSGA improves individual metrics in the evolutionary process and prevents an imbalance in the population diversity, by introducing a Hamming distance when initializing the population. As a result, the initial population covers the entire solution space. The theorem of the significance attribute employs a constraint condition. Thus, we assign a

value equal to one to the attribute reduction set position, according to the conditional risk features with the largest attribute dependence. We decide that the termination condition would reach 200 iterations, in order to analyze the algorithm diversity directly.

When using the genetic algorithm, we first consider the genetic representation and design of the fitness function. The former method uses a fixed n-bit binary series if the actual characteristic of attribute reduction is considered, in line with [31]. This study's encoding length is the number of conditional attributes in the decision table and the conditional risk attributes' room *C* is mapped into an individual room when each bit corresponds to a conditional risk attribute. The latter relies on two aspects—the number of and ability to classify attributes. Both dimensions relate to the decision-making attribute set, as follows:

$$\min F(B) = p_1 \frac{|POS_C(D)|}{|U|} + p_2 \left( \frac{|C| - |B|}{|C|} \right) + p_3 \, \overline{S}(B). \tag{1}$$

The function consists of three parts. The first part represents the classification ability. A reduction occurs when $k = 1$. The second part shows the reduction rate. For example, the fewer attributes in the attribute subset *B* indicate that *B* is the minimum reduction with a high reduction rate. The third part represents the weight factor, which increases the efficiency of the algorithm. The three elements are dynamically adjusted during the algorithm evolution. In this study, the adaptive factors represent the adjustment parameters used to ensure the accuracy of the reduction results with minimum reduction.

Based on the definition of RST approximation, the reduction attribute sets of RSGA are obtained mathematically as $Red(B) \subseteq A$, where $Red(B)$ is the reduced set comprising a set of attributes *B*. Then, the RSGA also produces more than a reduced attribute set. As the final output, we deploy the core of RST. The core attribute set represents intersections of all the reduction attribute sets, which is the most important attribute set for decision-making. Therefore, the core attribute set is mathematically defined as $Core(B) = \cap Red(B)$.

*3.3. The Framework of the Threat Utility-Based Hybrid Rough Set Design*
3.3.1. Description of Threat Utility Problems

There is a need to address the limitations of the threat utility of PSCRD. Some of the limitations are solving the MCDM problem with multiple expert distortion, the interdependency among several seaports risk factors, and deriving a ranking of alternatives [12]. The first problem can be solved by generating a trade-off by integrating the subordinate ranking sets. However, the difficulty is to integrate the ordinal information from the decision table without losing the decision-maker contexts. The second problem lies in obtaining the tendency attributes by reducing the superfluous features without losing the classification ability of the approach. The third problem is concerned with how to select the optimal alternative that has the biggest probability to achieve the threat utility of all the given criteria. To solve the three aforementioned issues, this study adopts the hybrid-conjoint approach with an adjusted MCDM problem.

In the given context, using the RGSA, we evaluate the experts' judgement on conditional seaport risk factors by referring to a design range in Table 2. Subsequently, we present the threat variables as latent variables, which contain several seaport risks factors. Second, we generate a risk profile using an orthogonal scheme. We evaluate the risk profiles as independent variables by aggregating the latent variables. Hence, the evaluation of the "design range" and RGSA help in solving the first MCDM problem. The aggregation of an alternative also helps in addressing the second and the third issues and in understanding the performance of the "system range" as input parameters of the PSCRD model.

**Table 2.** Indices of seaport risk assessment.

| Dimensional Threats | Definition Scale: Conditional Seaport Risk Evaluation (Code Form) |
|---|---|
| Planning process threat ($A_1$) | |
| Infrastructure threat ($A_2$) | Highest level: Loss of ability to perform operations and/or meet customer requirements. |
| Seaport service process threat ($A_3$) | High level: Temporary interruption or discontinuity in normal operations, delivery of goods, and/or services to customers. |
| Distribution process threat ($A_4$) | Medium level: Postponement in regular operations, plans and schedules, conveyance of products, and/or service to customers. |
| Relationship process threat ($A_5$) | Low level: Deviation in transportation plans, costs, common operations, timetables, quality, measure of conveyed merchandise |
| Nuclear-enterprise financial threat ($A_6$) | (product), and/or services to customers. |
| Monetary threat ($A_7$) | Lowest level: Operations remain unaffected or experience a |
| Location threat ($A_8$) | negligible effect |
| Security threat ($A_9$) | |
| Environmental threat ($A_{10}$) | |

Thus, we create a new decision table comprising a finite set of conditional seaport risk factors (criteria), $C = \{c_1, c_2, \ldots, c_n\}$, with a weight vector $W = \{w_1, w_2, w_n\}^T$. We invite a group of experts $E = \{e_1, e_2, \ldots, e_q\}$ to assess these criteria. Each expert, $e_q$, needs to determine the functional requirement (design range) of each criterion. Subsequently, we propose an aggregation for expert evaluations on the alternative denoted as $AR = \{ar_1, ar_2, \ldots, ar_J\}$. The "design range" has the same expression type as the "system range" corresponding to everyone. Hence, if an expert evaluates the "design range" of a criterion as a numerical number (linguistic term risk level), then the "system range" of the alternative on this criterion will be also expressed in the numerical form. The new aggregation of the individual decision matrix, $AR_{m \times n}^{(q)}$, can be established as follows:

$$AR_{m \times n}^{(e_q)} = \begin{bmatrix} \frac{\eta_{11}-\omega_1}{\mu_{11}} & \frac{\eta_{12}-\omega_2}{\mu_{12}} & \frac{\eta_{1j}-\omega_i}{\mu_{1j}} \\ \frac{\eta_{21}-\omega_1}{\mu_{21}} & \frac{\eta_{22}-\omega_2}{\mu_{22}} & \frac{\eta_{2j}-\omega_i}{\mu_{2j}} \\ \vdots & \ddots & \vdots \\ \frac{\eta_{i1}-\omega_1}{\mu_{i1}} & \frac{\eta_{i2}-\omega_2}{\mu_{i2}} & \frac{\eta_{ij}-\omega_i}{\mu_{ij}} \end{bmatrix} \tag{2}$$

The RSGA deploys core attributes with a high significance degree. The group of core attribute sets is defined as a core attribute space, $Core^a(B)$, where the core attribute $a$ is in the subset $B$, according to the running algorithm. Thus, we deploy an importance degree to estimate the relative importance of the core attribute space as follows:

$$\Lambda_j = \sum_{Core^a(B)} \frac{\gamma_j(B, D)}{U}, \tag{3}$$

$$\omega_j = \frac{\Lambda_j}{\sum \Lambda_j}. \tag{4}$$

The results of RSGA that come along with the core attributes represent the dependency degree for each core attribute. We assume that the ratio between the dependency score and the total observation represent the average of the dependency degree (importance degree) in Equation (3). We propose the weight indicator in Equation (4) to estimate the relative risk importance of the dependency degree toward the threat variables. Thus, all notation such as sets, indices, parameters, and decision variables used in this study are cited in the Appendix A.

### 3.3.2. Orthogonal Design

In this study, we adopt a hybrid-conjoint approach, in which the given risk combination of core attributes is computed using an expert aggregation based on a set of orthogonal designs. Subsequently, we use the utility function to understand the utility in relation to the risk level. The main concept in this orthogonal design is to create a set of orthogonal profiles, subdivide them, and assign them to each subject in a subgroup of people [32]. Each conditional seaport risk profile receives the same amounts of replications due to the overall administration. Let $ar$ = number of risk profile in the orthogonal design, $\Gamma$ = replications for each profile, $\tau$ = number of profiles administered to any one expert, $\beta$ = number of blocks of profile (each block is administered to one expert in the study). Then, in the balanced incomplete block designs, the following conditions hold. First, each profile appears once, at most, in a block. Second, each profile appears exactly $r$ times in administration. Finally, each pair of profile occurs exactly $l$ times together. In such a way, the following conditions hold among the parameters of design:

$$ar.\Gamma = \beta.\tau \text{ and } l(ar - 1) = \Gamma(\tau - 1) \tag{5}$$

After deploying the profile of seaport risk factors, we compute the experts' responses according to Equation (2) with the following equation.

$$DR = \sum_{J=1}^{J_q} ar_j^{(q)} = \sum \frac{\eta_{ij}^q - \omega_i}{\mu_{ij}^q} \tag{6}$$

### 3.3.3. Weight of Observations

This section solves the problems of how to determine the aggregate performance of individual expert evaluations and how to combine the collective information content determined by the RSGA method with the collective information to drive the ranking sets of experts' observations. Thus, the weighted aggregated sum product assessment (WASPAS) initialized by [33] is modified to determine the total importance of observation for aggregating experts' responses in this MCDM problem, which is mathematically defined as follows:

$$Q_l = \lambda \sum_{j=1}^{m} ar_j^{(q)} \omega_{ij} + (1 - \lambda) \prod_{j=1}^{m} \left( ar_j^{(q)} \right)^{\omega_{ij}}, \ \lambda = 0, \dots, 1 \tag{7}$$

### 3.3.4. Utility Representations Forms

This study makes two considerations in order to obtain the threat function of a risk combination (profile), such as effective profile and aggregation risk levels of a profile. The first consideration is explained in the orthogonal design. The second consideration presents a discretization of the linguistic term set ($V_a$), which is a set of possible linguistic terms of the linguistic variables. Subsequently, we denote the continuous-valued linguistic term as $Z = [z_{0.1}, z_1]$, where $z_{0.1}$ and $z_1$ represent the lowest and highest risks, respectively [34]. In this study, the threat utility follows the assumption that the sum of satisfaction is achieved by reducing the risk level. In other words, the experts always follow the principle of the minimum cost. Thus, the marginal utility can be presented in the following Equation (8) as the ratio between the utility and the level of risk factors for each risk factor.

$$MU_Q(m) = \frac{\Delta U_q}{\Delta Z} \tag{8}$$

Furthermore, $V_a$ is nominally scaled. Assume that $A_{mn}$ has $L_n$ levels. Let $D_{mLn-1}$ be the dummy variables for the $m$-th attribute. Then, the utility function is:

$$U(m) = \beta_0 + \sum \beta_{ij} D_{iL_j - 1} \tag{9}$$

In order to filter the effect of the noise factors affecting the result of the conjoint measurement, let us suppose a new information system, in which the response $Y$ (e.g., a disruption level) aggregated by $q$ respondents to risk profiles composed of $m$ threat attributes can be modelled by a multiple linear regression:

$$\overline{Y} = \overline{\overline{A}} \bullet \overline{\beta} + \overline{\varepsilon} \tag{10}$$

where

$$\overline{Y} = \begin{bmatrix} U_1 \\ U_2 \\ \vdots \\ U_l \end{bmatrix}, \ \overline{\overline{A}} = \begin{bmatrix} 1 & a_{11} & a_{12} & \cdots & a_{1m} \\ 1 & a_{21} & a_{22} & \cdots & a_{2m} \\ & \vdots & \vdots & & \vdots \\ 1 & a_{q1} & a_{q2} & \cdots & a_{qm} \end{bmatrix}, \ \overline{\beta} = \begin{bmatrix} \beta_1 \\ \beta_2 \\ \vdots \\ \beta_m \end{bmatrix}, \ \overline{\varepsilon} = \begin{bmatrix} \varepsilon_1 \\ \varepsilon_2 \\ \vdots \\ \varepsilon_l \end{bmatrix} \tag{11}$$

$a_{lm}$ is the value of the $m$-th attribute ($m = 1, 2, \ldots, M$) in the profile aggregated from the $l$-th expert evaluation, $\overline{\beta}$ is a vector of risk parameters, and $\overline{\varepsilon}$ is a vector of random variables modelling the measurement error.

Let us introduce the Equation (6) as the risk multiplicative coefficient in the matrix $\overline{\overline{A}}$ mathematically as:

$$\overline{\overline{A}}_{new} = \begin{bmatrix} 1 & ar_1^{(1)}a_{11} & ar_2^{(1)}a_{12} & \cdots & ar_j^{(1)}a_{1m} \\ 1 & ar_1^{(2)}a_{21} & ar_2^{(2)}a_{22} & & ar_j^{(2)}a_{2m} \\ & \vdots & & \ddots & \vdots \\ 1 & ar_j^{(q)}a_{q1} & ar_j^{(q)}a_{q2} & \cdots & ar_j^{(q)}a_{qm} \end{bmatrix} \tag{12}$$

Concerning the results, they affected the model in the Equation (10). In the case in which $w_{qj} = w_j$ ($\forall q = 1, 2, \ldots, l \ \forall j = 1, 2, \ldots, m$)—the multiplicative coefficients are different for each attribute but common for all the respondents—$\overline{\beta}_{new}$ is related to $\overline{\beta}$:

$$\overline{\beta}_{new} = \left[ \beta_0, \frac{\beta_1}{ar_1^{(1)}}, \ldots, \frac{\beta_j}{ar_j^{(q)}} \right]^T \tag{13}$$

Equation (13) is a purely formal passage. It provides a possibility of introducing the relative weight of importance according to Equation (7) in the model. In fact, we observe the model parameters increase proportionally to the relative weight importance $w_j$ given by the respondents to the $m$-th attribute. Thus, by introducing the multiplicative coefficient, we obtain the following formula:

$$\overline{\beta}_{new} = W \circ \overline{\beta}, \tag{14}$$

where $w \in Q_l$, $W = \{w_1, w_2, w_n\}^T$, and the symbol 'o' denotes the Hadamard product between vectors $W$ and $\overline{\beta}$—$\overline{\beta}_{new}$ is obtained by multiplying element-by-element the vectors $W$ and $\overline{\beta}$.

Ultimately, the framework of the threat utility-based hybrid rough set design is depicted in Figure 1.

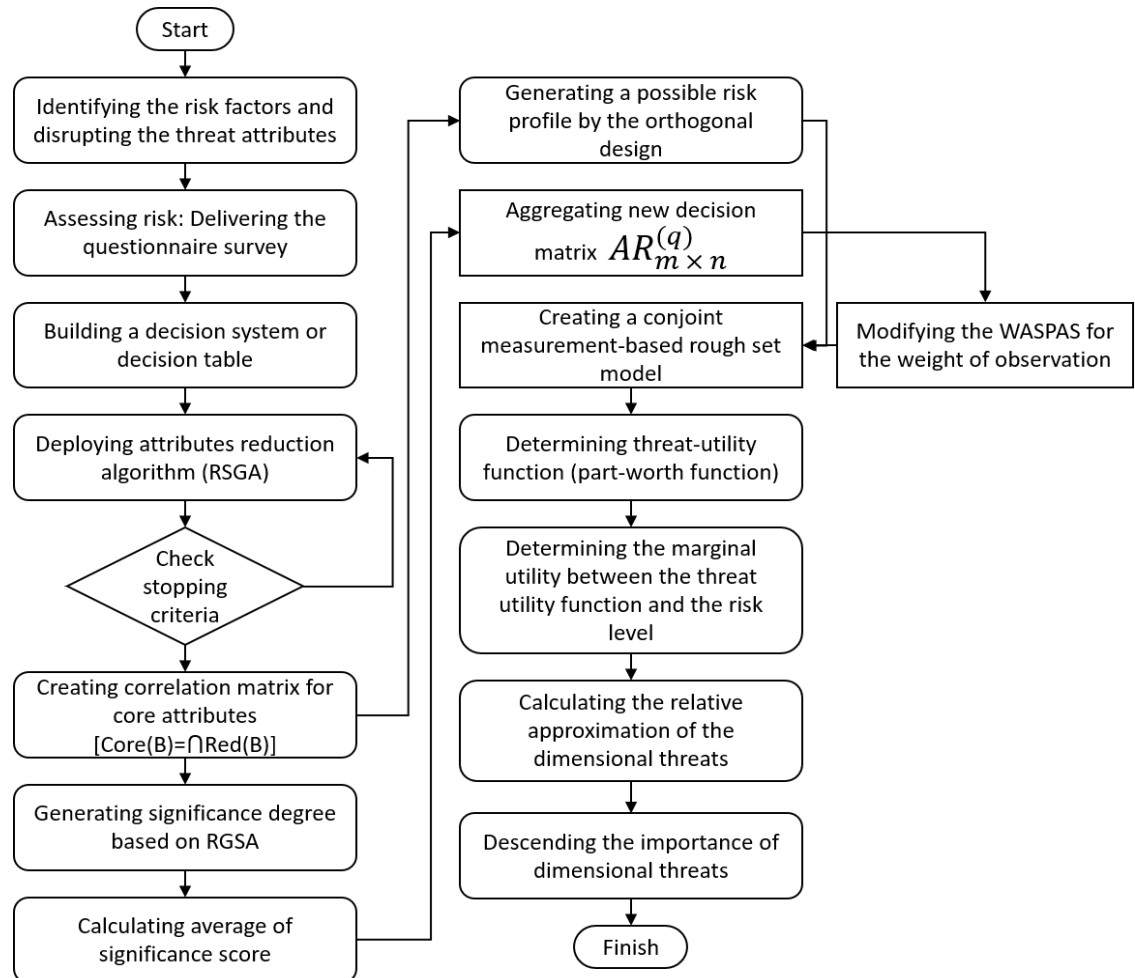

**Figure 1.** Flowchart of the threat utility.

## 4. An Empirical Analysis of Threat Utility Study

### 4.1. Data Selection and Coding the Attributes

We select Indonesia as a case study due to its complexity, as mentioned in the literature. Ref. [15] shows at least 9755 cases of disruption management. Typical causes of disruption are "disobedience" in terms of operational rules, administrative regulations, and ministry decrees; weakness in the control systems, such as accounting and financial control; and policy. Both directly and indirectly, these factors relate to the export and import trade, as well as supply chain continuity and accidents with victims (either infrastructure or people). These phenomena reduce the seaport risk predictability.

We mainly source the dimensional threats and 61 risk attributes' results based on an extensive examination of the literature—particularly [1,17,21,35]—and discussion with some experts. We identify several indexes for capturing the different perspectives of domain experts. They are categorized into dimensional threat groups as top events with conditional seaport risk attributes, as shown in Figure 2. We design this classification of threats to reflect on the responsiveness to the threat in terms of the seaport supervision level. We define the scope of actions that may be anticipated as related to the conditional risks. Each attribute is coded, which means that each attribute is assigned to the appropriate numerical value, as depicted in Table 2.

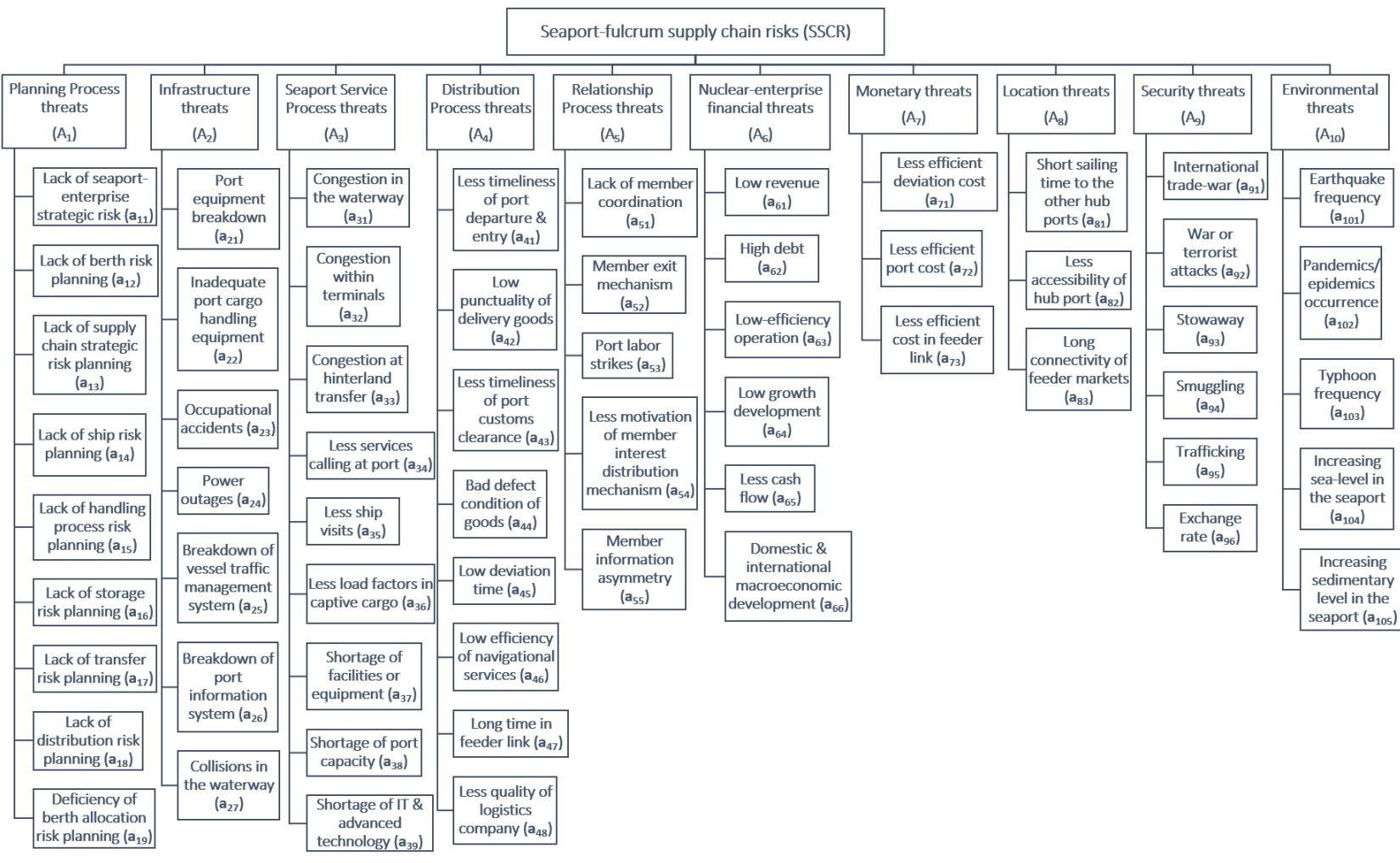

**Figure 2.** Supply chain disruption events under seaport risk factors.

### 4.2. Data Collection Procedures

We employ an online questionnaire survey, face-to-face interviews, and focused group discussion. We use a stratified random sampling technique with Slovin's formula to choose the population sample [36]. The data collection period spans from January to August 2021. This study classifies the port-centric supply chain stakeholders into three main stakeholder categories that significantly impact the supply chain issues—seaport managers (10%), operators (40%), and users (50%). The seaport-manager is a port authority figure who works for a government agency or a government-owned company. The seaport-operator is in control of the seaport firm's operating procedure, which includes containers and non-containerized commodities such as automobiles, liquids, and dry bulk. Finally, seaport-users are stakeholders that collaborate with seaport operators and have a direct interest in the goods moved via seaports such as cargo owners, freight forwarders, ship owners, and ship management companies.

We collect 153 data units, in accordance with the planned collection target of 150 data units. Table 3 presents the demographics of these respondents. We examine the potential of supply chain disruption in relation to the latent variables. We obtain the aggregating latent variables from several conditional risk attributes and decisional risk factors evaluated by a five-level ordinal scale, which indicates the risks to supply chain continuity, shown in Table 2. Both risk attributes and factors are evaluated by the port-centric supply chain stakeholders. The study adapts a rough set model through a questionnaire survey to find the central dependency among the many factors, corresponding to Figure 2. The proposed algorithm generates a reduction attribute set that helps obtain a core attribute set. The set of core attributes is crucial for understanding the central seaport-focal supply chain risk tendency. Figure 1 illustrates the entire procedure.

**Table 3.** Demographic information of respondents.

| No. | Demographic Factor | | Percentage |
|---|---|---|---|
| 1. | Gender | Male | 70% |
| | | Female | 30% |
| 2. | Object of research | Seaport manager | 11% |
| | | Seaport operator | 43% |
| | | Seaport user | 47% |
| 3. | Work duration | Below 5 years | 4% |
| | | Between 5–10 years | 39% |
| | | Over 10 years | 57% |
| 4. | Educational background | Diploma | 10% |
| | | Bachelor | 61% |
| | | Master's | 27% |
| | | Doctoral | 2% |

### 4.3. Reliability Test

To initiate the RSGA model, we conduct the initial test of primary data. This helps us to check the reliability of the dataset after collection. We test 61 conditional seaport risk factors and a decisional factor from 153 responses. Table 4 shows the results of the reliability test. According to the Cronbach's alpha, we find the dataset "very reliable" because these values are more than 0.80 as an input for the rough set-based genetics algorithm.

After the test of the primary datasets of the questionnaire survey, we run the RGSA to obtain the core attributes as the central tendency of PSCRD. Subsequently, we conduct the reliability test to check the aggregation of seaport risk profiles, corresponding to the

orthogonal design. Specifically, we check the reliability of the dataset of 32 risk profile, as presented in Table 4 below.

**Table 4.** Result of reliability test.

| Case Processing Summary | | Number of Responses | Percentage |
|---|---|---|---|
| | Valid | 153 | 100% |
| Cases | Excluded [a] | 0 | 0.0 |
| | Total | 153 | 100% |
| Reliability results for questionnaire survey | | | |
| Cronbach's alpha | | 0.882 | 88.2% |
| Number of features | | 62 | 100% |
| Reliability results for the aggregation of seaport risk alternative | | | |
| Cronbach's alpha | | 0.991 | 99.1% |
| Cronbach's alpha based on standardized items | | 0.992 | 99.2% |
| Number of features | | 32 | 100% |

Note: [a] Listwise deletion based on all variables in the procedure.

### 4.4. Evaluation of Core Attributes and Orthogonal Design

After receiving the core attribute from the RSGA process, we run the computation corresponding to Equations (3) and (4) to provide the preference order. The preference order prioritizes the latent factors in the threat utility function [37]. The order also helps us to understand the utility value and the discretization of risk levels. The RSGA highlights 24 seaport risk criteria, each corresponding to ten threat factors, based on Table 5. The orthogonal design of the seaport risk criteria helps in estimating the threat utility function. The error term in Equation (10) is essentially the same as the random part of the utility function. For this analysis, we define the dummy variables $D_{mn} = J - 1$ for the ten latent variables as follows:

$$D_{mn} = \begin{cases} 1, & \text{if } m \text{ latent variable is n risk criteria} \\ 0, & \text{otherwise} \end{cases} \tag{15}$$

Appendix B presents the orthogonal design of the seaport risk criteria.

### 4.5. Threat Utility of PSCRD

4.5.1. Part-Worth Utility and Its Threat Utility Function

In this study, the theories of threat utility develop in line with two trends—the topological-set and probabilistic trends. The topological-set trend (conditional seaport risk tendency) assumes the non-measurability of the utility. Using RGSA, we obtain the set trend emerging from the expert evaluation. The probability trend can be defined by referring to the available risk profiles, corresponding to the risk tendency set. Furthermore, by introducing Equations (2) and (7) and by the monotonic arrangement of variants (seaport risk profiles) in descending order, we determine the direction of the seaport risk criteria [32]. Given this, we define the threat utility as the sum of the disruption level by the seaport risk factor that influences the satisfaction of port-centric supply chain continuity. Meanwhile, the marginal utility in Equation (8) presents the disrupted satisfaction, according to the setup of the risk level. Both quantifications are carried out by the tendency expressed with the part-worth utility function in the following Table 6.



**Table 5.** RSGA score of the port-centric supply chain risk.

| No. | Dimensional Threats | Attributes | Importance Degree ($\Lambda_{ij}$) | Weight ($\omega_{ij}$) |
|---|---|---|---|---|
| 1 | Planning Process threat ($A_1$) | $a_{14}$ | 0.023 | 0.048 |
| 2 | | $a_{15}$ | 0.021 | 0.042 |
| 3 | | $a_{16}$ | 0.015 | 0.031 |
| 4 | | $a_{18}$ | 0.014 | 0.029 |
| 5 | Infrastructure threat ($A_2$) | $a_{25}$ | 0.073 | 0.149 |
| 6 | | $a_{26}$ | 0.019 | 0.038 |
| 7 | Seaport Service Process threat ($A_3$) | $a_{31}$ | 0.018 | 0.038 |
| 8 | | $a_{33}$ | 0.037 | 0.076 |
| 9 | | $a_{34}$ | 0.014 | 0.028 |
| 10 | | $a_{38}$ | 0.016 | 0.033 |
| 11 | | $a_{39}$ | 0.025 | 0.050 |
| 12 | Distribution Process threat ($A_4$) | $a_{42}$ | 0.015 | 0.030 |
| 13 | | $a_{43}$ | 0.023 | 0.046 |
| 14 | Relationship Process threat ($A_5$) | $a_{54}$ | 0.092 | 0.188 |
| 15 | Nuclear-enterprise financial threat ($A_6$) | $a_{61}$ | 0.015 | 0.031 |
| 16 | | $a_{64}$ | 0.020 | 0.041 |
| 17 | Monetary threat ($A_7$) | $a_{73}$ | 0.035 | 0.072 |
| 18 | Location threat ($A_8$) | $a_{81}$ | 0.022 | 0.045 |
| 19 | Security threat ($A_9$) | $a_{91}$ | 0.022 | 0.044 |
| 20 | | $a_{94}$ | 0.021 | 0.043 |
| 21 | | $a_{95}$ | 0.043 | 0.087 |
| 22 | Environmental threat ($A_{10}$) | $a_{101}$ | 0.021 | 0.043 |
| 23 | | $a_{102}$ | 0.032 | 0.066 |
| 24 | | $a_{105}$ | 0.020 | 0.041 |

**Table 6.** Part-worth utility function for seaport risk model.

| No. | Threat Variables | Part-Worth (Partial) Utility Function |
|---|---|---|
| 1. | Planning process threat ($A_1$) | $U(A_1) = -0.051a_{14} + 0.046a_{15} + 0.054a_{16} - 0.049a_{18}$ |
| 2. | Infrastructure threat ($A_2$) | $U(A_2) = 0.032a_{25} - 0.032a_{26}$ |
| 3. | Seaport service process threat ($A_3$) | $U(A_3) = 0.009a_{31} - 0.062a_{33} - 0.026a_{34} + 0.008a_{38} + 0.072a_{39}$ |
| 4. | Distribution process threat ($A_4$) | $U(A_4) = 0.029a_{42} - 0.029a_{43}$ |
| 5. | Relationship process threat ($A_5$) | $U(A_5) = 0.371a_{54} - 0.371a_{5N}$ |
| 6. | Nuclear-enterprise financial threat ($A_6$) | $U(A_6) = 0.024a_{61} - 0.024a_{64}$ |
| 7. | Monetary threat ($A_7$) | $U(A_7) = 0.331a_{73} - 0.331a_{7N}$ |
| 8. | Location threat ($A_8$) | $U(A_8) = 0.248a_{81} - 0.248a_{8N}$ |
| 9. | Security threat ($A_9$) | $U(A_9) = 0.012a_{91} + 0.016a_{94} - 0.028a_{95}$ |
| 10. | Environmental threat ($A_{10}$) | $U(A_{10}) = -0.002a_{101} + 0.022a_{102} - 0.02a_{105}$ |

In Table 6, in the part-worth utility function, each seaport risk factor for each threat variable is affected in proportion to the disrupted utility port-centric supply chain continuity. For example, the disrupted utility of the port-centric supply chain, due to the planning

process threat ($A_1$), will increase by 1% only if the lack of ship risk planning ($a_{14}$) and the lack of distribution risk planning ($a_{18}$) decline by as much as −5%, while the lack of handling process risk planning ($a_{15}$) and the lack of storage risk planning ($a_{16}$) increase by 5%. Furthermore, the "N" factor in $A_5$, $A_7$, and $A_8$ means that other port-centric supply chain risk variables are in proportion to factors $a_{54}$, $a_{73}$, and $a_{81}$ causing the PSCRD. This factor occurs as a result of the dummy coding [32]. Figure 3 below illustrates the whole part-worth function on seaport risk criteria factors.

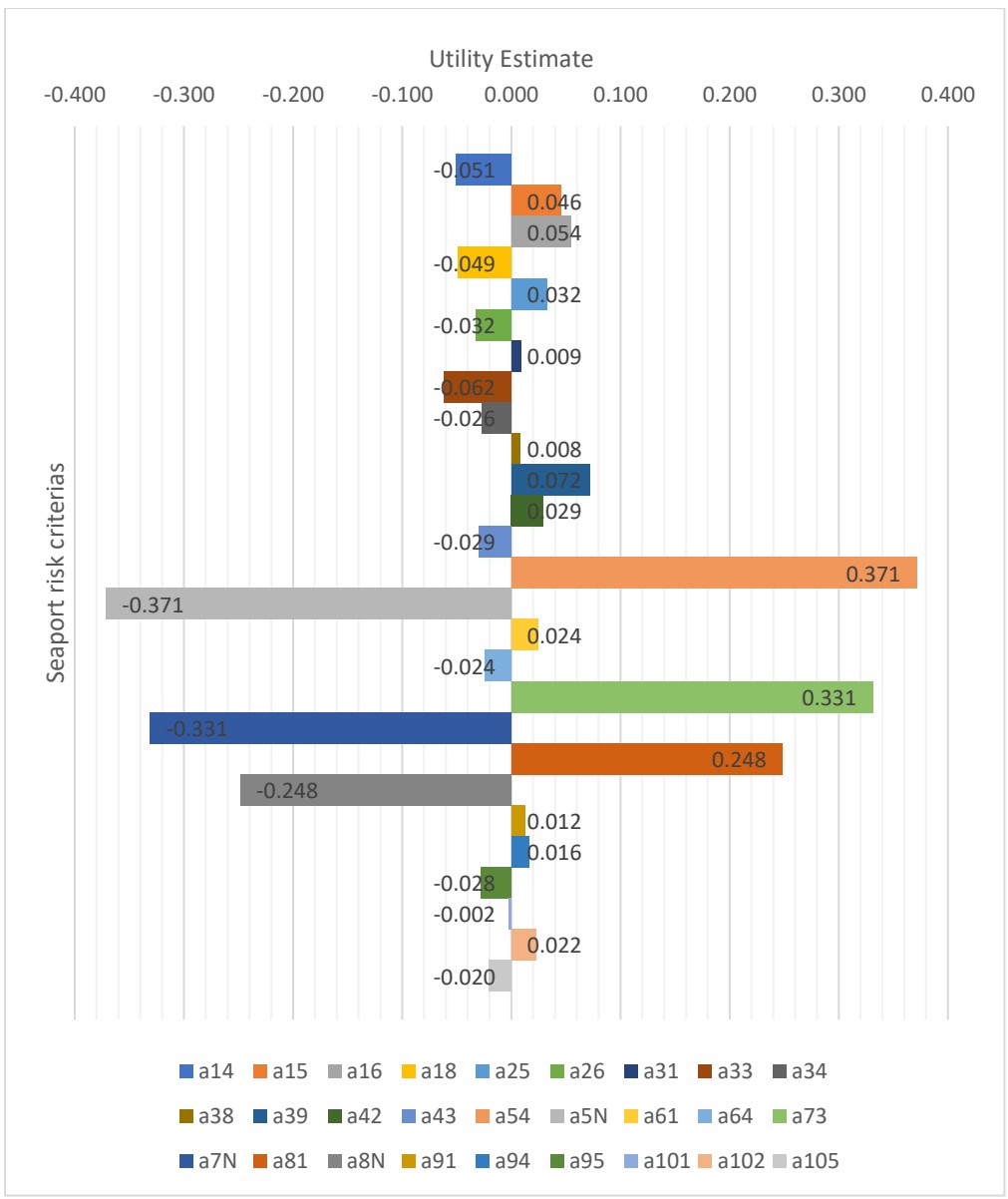

**Figure 3.** Conditional seaport risks ratio.

### 4.5.2. Importance of Dimensional Threats

In the PSCRD context, the seaport service process should be optimized as a priority in order to reduce the disruption. The factor disrupts utility by as much as 32%, followed by the relationship process threat ($A_5$) and planning process threat ($A_1$). Hence, these latter two threats assume the second and third priorities, respectively. However, the infrastructure breakdown ($A_2$) poses less threat of PSCRD at 9.9%. Figure 4 depicts the potential threats. Table 7 and Equation (14) provide the background risk, and thereby help

the seaport manager, operator, and user to determine their resilience path toward these potential threats.

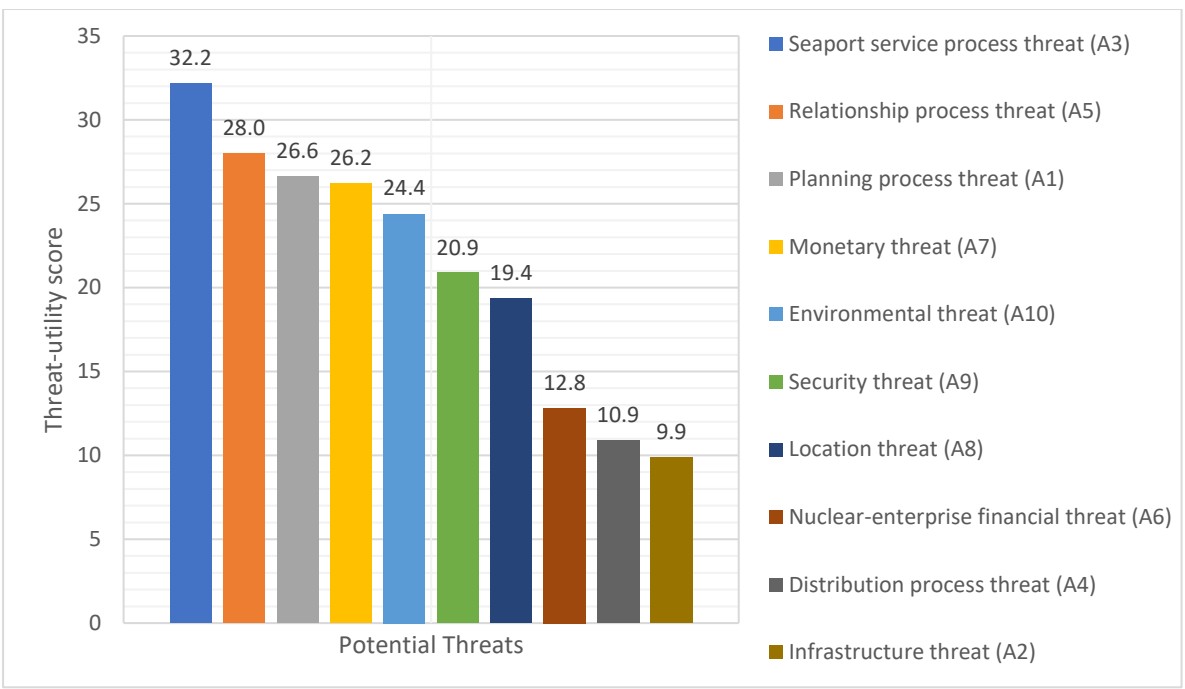

**Figure 4.** Potential threats of the PSCRD in Indonesia.

*4.6. Results and Discussion*

In order to determine the impact of each potential threat factor on the satisfaction of port-centric supply chain continuity, we present the partial utility corresponding to the conjoint measurement in Table 6. The threat utility models (partial utility) examine the correlation between two or more seaport risk criteria. The threat utility analysis starts with the 24 conditional seaport risk factors extracted using the RSGA. The algorithm generates 24 risk factors from 153 evaluations of experts from three stakeholder categories—seaport manager, operator, and user. Based on the results obtained from the proposed model analysis, as shown in Table 7, the potential threats to disrupt port-centric supply chain continuity originate from the 24 seaport risk factors. Referring to Equation (11), several seaport risk criteria are independent variables that affect the utility of the threat as dependent variables. This is depicted in Figure 3. In the final output, the threat factors are introduced as latent variables to estimate the disruption level of the PSCRD model.

Table 6 describes the empirical equation from the results of the partial utility of each dependent variable, based on the distribution of PSCRD entities. The formula in the function column shows predictions of the seaport risk factors. As a result, the validation model of the final output in Equation (16) shows the weighted estimation of experts' judgement in C and D. We conduct the partial utility using the stepwise method, which allows the model containing significant predictor values to be obtained from the expert model individually in Table 7. A higher $R^2$ (maximum value 1.00) indicates that the individual model has high predictive ability. Hence, the PSCRD model is considered good to explain the empirical analysis of the port-centric supply chain utility in Equation (16).

$$Y_{mn} = 1.33 + 26.7U(A_1) + 9.9U(A_2) + 32.2U(A_3) + 10.9U(A_4) + 28.0U(A_5) + 12.8U(A_6) + 26.2U(A_7) \\ + 19.4U(A_8) + 20.9U(A_9) + 24.4U(A_{10}) \tag{16}$$

**Table 7.** Estimated utility of PSCRD factors.

| Risk Factors | Estimated Utility | | | | | | |
|---|---|---|---|---|---|---|---|
| | Overall | $e_1$ | $e_2$ | $e_3$ | $e_4$ | ~ | $e_{153}$ |
| Constant | 1.33 | 0.104 | 0.162 | 0.082 | 0.035 | ~ | 0.137 |
| $a_{14}$ | −0.05 | 0.155 | 0.242 | 0.123 | 0.052 | ~ | 0.204 |
| $a_{15}$ | 0.05 | 0.155 | 0.242 | 0.123 | 0.052 | ~ | 0.204 |
| $a_{16}$ | 0.05 | 0.155 | 0.242 | 0.123 | 0.052 | ~ | 0.204 |
| $a_{18}$ | −0.05 | 0.155 | 0.242 | 0.123 | 0.052 | ~ | 0.204 |
| $a_{25}$ | 0.03 | 0.090 | 0.140 | 0.071 | 0.030 | ~ | 0.118 |
| $a_{26}$ | −0.03 | 0.090 | 0.140 | 0.071 | 0.030 | ~ | 0.118 |
| $a_{31}$ | 0.01 | 0.168 | 0.262 | 0.133 | 0.056 | ~ | 0.221 |
| $a_{33}$ | −0.06 | 0.168 | 0.262 | 0.133 | 0.056 | ~ | 0.221 |
| $a_{34}$ | −0.03 | 0.168 | 0.262 | 0.133 | 0.056 | ~ | 0.221 |
| $a_{38}$ | 0.01 | 0.218 | 0.340 | 0.173 | 0.073 | ~ | 0.287 |
| $a_{39}$ | 0.07 | 0.218 | 0.340 | 0.173 | 0.073 | ~ | 0.287 |
| $a_{42}$ | 0.03 | 0.090 | 0.140 | 0.071 | 0.030 | ~ | 0.118 |
| $a_{43}$ | −0.03 | 0.090 | 0.140 | 0.071 | 0.030 | ~ | 0.118 |
| $a_{54}$ | 0.37 | 0.090 | 0.140 | 0.071 | 0.030 | ~ | 0.118 |
| $a_{5N}$ | −0.37 | 0.090 | 0.140 | 0.071 | 0.030 | ~ | 0.118 |
| $a_{61}$ | 0.02 | 0.090 | 0.140 | 0.071 | 0.030 | ~ | 0.118 |
| $a_{64}$ | −0.02 | 0.090 | 0.140 | 0.071 | 0.030 | ~ | 0.118 |
| $a_{73}$ | 0.33 | 0.090 | 0.140 | 0.071 | 0.030 | ~ | 0.118 |
| $a_{7N}$ | −0.33 | 0.090 | 0.140 | 0.071 | 0.030 | ~ | 0.118 |
| $a_{81}$ | 0.25 | 0.090 | 0.140 | 0.071 | 0.030 | ~ | 0.118 |
| $a_{8N}$ | −0.25 | 0.090 | 0.140 | 0.071 | 0.030 | ~ | 0.118 |
| $a_{91}$ | 0.01 | 0.120 | 0.186 | 0.095 | 0.040 | ~ | 0.157 |
| $a_{94}$ | 0.02 | 0.140 | 0.219 | 0.111 | 0.047 | ~ | 0.184 |
| $a_{95}$ | −0.03 | 0.140 | 0.219 | 0.111 | 0.047 | ~ | 0.184 |
| $a_{101}$ | 0.00 | 0.120 | 0.186 | 0.095 | 0.040 | ~ | 0.157 |
| $a_{102}$ | 0.02 | 0.140 | 0.219 | 0.111 | 0.047 | ~ | 0.184 |
| $a_{105}$ | −0.02 | 0.140 | 0.219 | 0.111 | 0.047 | ~ | 0.184 |
| $R^2$ coefficients | 1.00 | 0.705 | 0.766 | 0.899 | 0.756 | ~ | 0.769 |

Figure 4 shows that, in the Indonesian context, the seaport service process ($A_3$) is the primary source of PSCRD. In other words, this tendency of PSCRD reduces the utility (satisfaction) by 32.2% in the 100% utility estimation. This is followed by the relationship process threat ($A_5$) and the planning process threat ($A_1$) with 28% and 26.6% utility reduction, respectively. The three threats that cause less PSCRD are infrastructure threats ($A_2$), distribution process threats ($A_4$), and nuclear-enterprise financial threats ($A_2$) with 9.9%, 10.9%, and 12.8% utilities, respectively.

Based on Figure 4, the highest potential threat to port-centric supply chain continuity originates from waterway congestion, congestion at the hinterland transfer, fewer services calling at port, a lack of port capacity, and a lack of IT and modern technology. The utility of these threats is estimated at about −6% to 7%. It means that these seaport risk factors have the potential to reduce the port-centric supply chain operation and reduce its utility to around −6% to 7%. In line with this, Ref. [38] examined some risk factors that directly

contribute to congestion within terminals and waterways as well as hinterland transfer, which affects 63.9% of the relative importance of seaport development, implementation, and supply chain operation at Indonesia's seaport. The second potential threat is the low motivation of distributors. This factor adversely impacts the relationship building process among the entities, at a reduced satisfaction level of 37%. For example, inefficient and inadequate marine security inspection at sea leads to operational delays and increases the liability under the contract of carriage. Overall, it leads to delayed delivery of goods [11].

The third potential threat is posed by the planning process. It is key to maintaining the continuity of the port-centric supply chain operation. Thus, it is essential to develop an integrated plan to ensure the resilience of the port-centric supply chain operation. The seaport risk criteria factors, such as the ship, handling process, storage, and distribution risks' planning, provide an estimated satisfaction of around −5% to 5%. This provides a reason to plan for resilience in relation to a PSCRD. In fact, Ref. [15] highlighted that the storage planning problem induces demurrage in the loading port and leads to a significant decline in the ship utility and an increase in its operational cost.

In terms of the risk level, the risk level Z is assumed. This level represents an escalation in threat utility from $z_{0.1}$ to the peak seaport risk level $z_1$ shown in Figure 5. By introducing Equation (8), we can understand that some of the 24 conditional seaport risks underwent a positive trend, while others experienced a negative trend. The positive trend of risk factors in Figure 5 contributes to an increase in the threat utility. It indicates that the conditional seaport risk factors can be predictable. It also shows an increasing trend in the port-centric supply chain operation and development, such as shipping activities and seaport activities, in the Indonesian context. However, it comprehensively needs attention from the port-centric supply chain stakeholders.

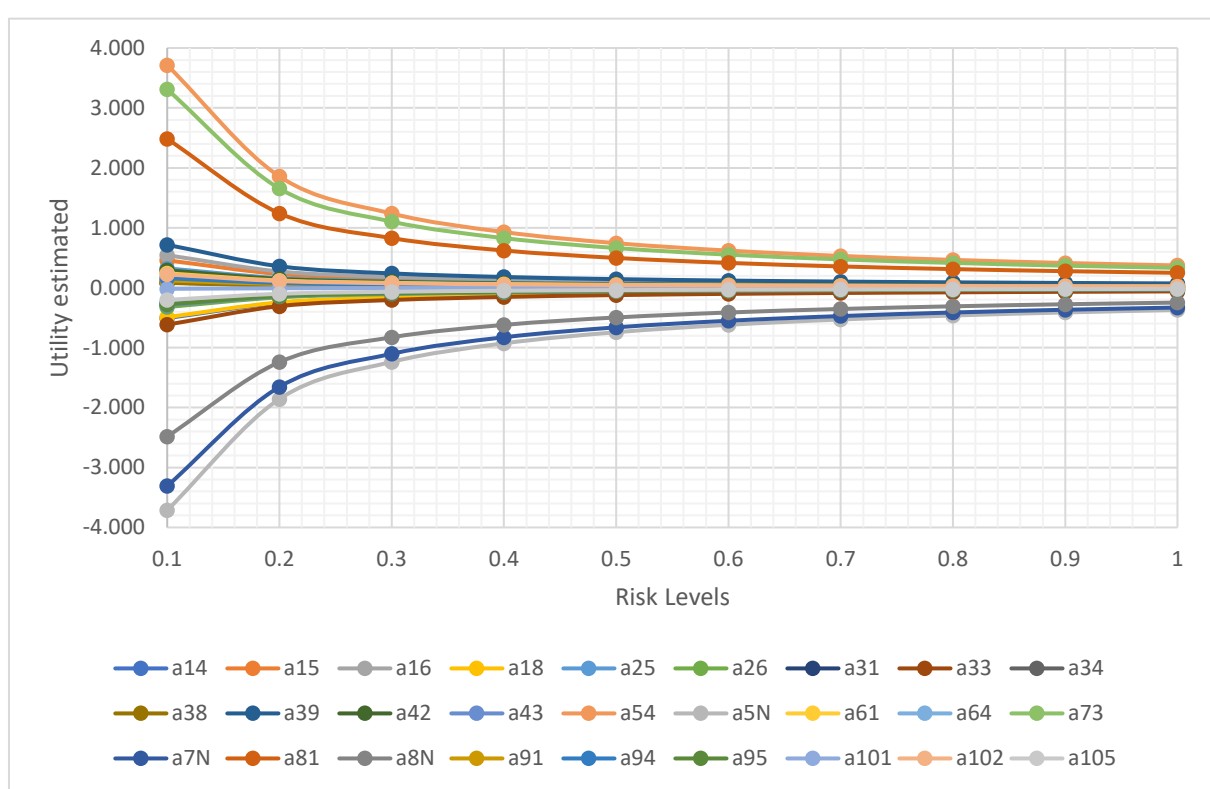

**Figure 5.** Trend of risk level toward seaport risk factors.

Meanwhile, the negative trend of the other risk factors in Figure 5 indicates that the threat utility is declining when the risk level is increasing. Hence, the conditional seaport risk factors significantly contribute to PSCRD and make the conditional seaport risk factors more unpredictable.

*4.7. Sensitivity Analysis*

In the proposed model, we check the performance of the aggregation in Equation (7) and Equation (11) by setting the accuracy parameter differently ($\lambda$). We generate $\lambda$ from zero to one to check the degree of sensitivity of the estimated threat utility toward the aggregation Equation (7). Hence, this sensitivity analysis relies on how to determine the aggregate optimal performance for each criterion (risk profile). In this step, the aggregate optimal performance is calculated as follows:

$$\mu_{iJ} = \begin{cases} \frac{x_{iJ} - x_{min_{iJ}}}{max_{iJ} - min_{iJ}}; & J \in \Omega_{normal} \\ \frac{min_{iJ}}{x_{iJ}}; & J \in \Omega_{min} \end{cases} \tag{17}$$

where $\Omega_{normal}$ denotes the set of average criteria, and $\Omega_{min}$ denotes the set of cost criteria.

Both criteria present a robustness parameter of the threat utility. The former refers to the initial condition before the robustness of risk, whereas the latter refers to an attempt to reduce the risk. In this sensitivity analysis, the output relies on the performance of the latent variables' aggregation. Given this, it is important to check the variations in the input variables of that model. The inputs in this MCDM model, with a hybrid conjoint approach, is the aggregation of Equation (6). Therefore, the variation in inputs based on the parameter of robustness is provided in Appendices C and D.

In Appendix E, the setup $\lambda$ is clearly affected by the quality of the input. A higher adjustment in $\lambda$ indicates the high quality of inputs. However, an increase in $\lambda$ in Appendix F does not always significantly improve the quality of the inputs. It must be noted that, irrespective of the adjustment level, the adjustment of $\lambda$ in the input slightly inclines or maintains the output value. If the model reflects the realistic situation, then an increment/decrement in the rate at which any input variable may occur would certainly result in a relative increment/decrement in the rate of the output node. Given this, an inference reasoning of the model is the key to generating the threat-utility knowledge.

## 5. Conclusions

In this study, the PSCRD model reveals that the 24 conditional seaport risk factors are the central tendency risk factors that play a key role in the port-centric supply chain operations. They impose ten potential threats that decrease the utility of the PSCRD model. In the Indonesian context, the key source of PSCRD is the ports' service procedure ($A_3$). In other words, the PSCRD tendency lowers utility (satisfaction) by up to 32.2% in a 100% utility assessment. It is followed by the relationship process threat ($A_5$) and the planning process threat ($A_1$), with utility estimates of 28% and 26.6%, respectively. Infrastructure risks ($A_2$), distribution process threats ($A_4$), and nuclear-enterprise finance threats ($A_2$) contribute less to PSCRD, with 9.9%, 10.9%, and 12.8% utilities, respectively.

Regarding the level of risk, the utility of port-centric supply chain exists when the port-centric supply chain risk peaks at the highest level. The highest level of port-centric supply chain factor is maintained at the utility level of 1–37%. The port-centric supply chain utility of an earthquake ($a_{101}$) is zero for the first time in the robustness test in Table 7. It means that the utility of this factor is zero in its first occurrence. This negative trend points out that port-centric supply chain entities must simultaneously focus on planning, implementation, and development directed toward enhancing port-centric supply chain resilience.

This study also provides a new holistic framework to analyze the PSCRDs that disrupt the port-centric supply chain operation, in relation to satisfaction. In relation to the threat utility, the study highlights the areas where each port-centric supply chain operation can lead to supply chain disruption related to the threat utility (potential threat). Given this, the study identifies a list of threat utility functions as indicators applicable to all the seaports in Indonesia. A limitation of this study is that the risk-level setting does not reflect the real situation of the dataset. Hence, future studies should consider another approach to

discretize the real risk level, referring to this dataset. Moreover, the port-centric supply chain stakeholders have different experiences and thoughts that are geographically diverse. Hence, it is possible to extract varied results from the RGSA. These variations might influence the findings on the threat utility.

**Author Contributions:** Conceptualization, M.R.D.B.; methodology, M.R.D.B.; software, M.R.D.B.; validation, M.R.D.B. and S.H.; formal analysis, M.R.D.B.; investigation, M.R.D.B.; resources, M.R.D.B.; data curation, M.R.D.B.; writing—original draft preparation, M.R.D.B.; writing—review and editing, M.R.D.B. and S.H.; visualization, M.R.D.B.; supervision, S.H. All authors have read and agreed to the published version of the manuscript.

**Funding:** This study was funded by the Indonesia Endowment Fund for Education (LPDP), Ministry of Finance, Republic of Indonesia (Kementerian Keuangan Republik Indonesia) through a scholarship scheme (Grant No. S-1975/LPDP.4/2022).

**Institutional Review Board Statement:** Not applicable.

**Informed Consent Statement:** Not applicable.

**Data Availability Statement:** The data is made referring the questionnaire survey from the Indonesian seaport stakeholders.

**Acknowledgments:** We appreciate the Indonesian Port's Authority, PT. Pelabuhan Indonesia (PELINDO), Persero—Regional 2, Regional 3, and Regional 4, PT. Kalla Lines, and other seaport-user companies for providing the data to our research. We are also grateful to the Ministry of Finance, Republic of Indonesia (Kementerian Keuangan Republik Indonesia) for supporting our research.

**Conflicts of Interest:** The authors declare no conflict of interest.

## Appendix A. Notation (Sets, Indices, Parameters, and Decision Variables)

- $U$ : Set of observation, indexed by $x \in U$.
- $A$ : Set of conditional risk factors, indexed by $a_{ij} \in A$ for all attributes ($C$ and $D$)
- $V_a$ : Set of risk magnitude or level for a-th attribute, indexed by $f(x,a) \in V_a$
- $I_B$ or $B$ : Set of indiscernible relation, indexed by $(x_1, x_2) \in I_B$
- $i$ : Threat factors, $i = 1, 2, \ldots, m$
- $j$ : Seaport risk factors, $j = 1, 2, \ldots, n$
- $J$ : Seaport risk profiles aggregated from the $q$-th expert, $J = 1, 2, \ldots, J_q$
- $q$ : Number of experts, $q = 1, 2, \ldots, l$
- $\eta$ : Expert evaluation in orthogonal design
- $\mu$ : Cost criteria normalization
- $Q$ : Ranking set of observation
- $\lambda$ : Initial criteria accuracy
- $\gamma$ : Dependency degree
- $S$ : Significant degree
- $p_1$ : Classification ability
- $p_2$ : Reduction rate
- $p_3$ : Correction factor
- $k_B$ : Dependency degree for each indiscernible relation $a_i \in I_B$
- $F_B$ : Attributes reduction set
- $\Lambda$ : Relative importance of the core attributes
- $\omega$ : Average of relative importance
- $\overline{S}$ : Significant degree indicator
- $DR$ : Design range of risk profile
- $Q_n$ : $J - 1$ = number of dummy variables for the $a$-th attribute
- $D_{mn}$ : The dummy variable for the $m$-th threat dimension of the $n$-th risk attribute (m = 1, 2, ..., $M_a$)
- $D_{mnJ}$ : Value of the dummy variable, $D_{mn}$ for the risk profile
- $U_a(m)$ : Utility function for the m-th dimension threat for q-th expert; m = 1, 2, ..., M

**Appendix B. Orthogonal Design of Seaport Risk Combinations (Risk Profile)**

| Numbers of Risk Profiles | Threat Factors (Latent Variables) | | | | | | | | | |
|---|---|---|---|---|---|---|---|---|---|---|
| | $A_1$ | $A_2$ | $A_3$ | $A_4$ | $A_5$ | $A_6$ | $A_7$ | $A_8$ | $A_9$ | $A_{10}$ |
| 1. | $a_{18}$ | $a_{25}$ | $a_{31}$ | $a_{42}$ | $a_{5N}$ | $a_{61}$ | $a_{7N}$ | $a_{8N}$ | $a_{91}$ | $a_{101}$ |
| 2. | $a_{18}$ | $a_{25}$ | $a_{39}$ | $a_{43}$ | $a_{54}$ | $a_{64}$ | $a_{7N}$ | $a_{8N}$ | $a_{91}$ | $a_{101}$ |
| 3. | $a_{18}$ | $a_{26}$ | $a_{31}$ | $a_{42}$ | $a_{5N}$ | $a_{61}$ | $a_{7N}$ | $a_{81}$ | $a_{94}$ | $a_{101}$ |
| 4. | $a_{14}$ | $a_{25}$ | $a_{31}$ | $a_{42}$ | $a_{54}$ | $a_{61}$ | $a_{73}$ | $a_{81}$ | $a_{91}$ | $a_{101}$ |
| 5. | $a_{18}$ | $a_{26}$ | $a_{33}$ | $a_{43}$ | $a_{54}$ | $a_{61}$ | $a_{73}$ | $a_{8N}$ | $a_{91}$ | $a_{102}$ |
| 6. | $a_{18}$ | $a_{26}$ | $a_{33}$ | $a_{43}$ | $a_{54}$ | $a_{61}$ | $a_{7N}$ | $a_{81}$ | $a_{95}$ | $a_{101}$ |
| 7. | $a_{16}$ | $a_{26}$ | $a_{38}$ | $a_{42}$ | $a_{5N}$ | $a_{61}$ | $a_{73}$ | $a_{81}$ | $a_{91}$ | $a_{101}$ |
| 8. | $a_{14}$ | $a_{26}$ | $a_{31}$ | $a_{42}$ | $a_{54}$ | $a_{61}$ | $a_{73}$ | $a_{8N}$ | $a_{95}$ | $a_{101}$ |
| 9. | $a_{14}$ | $a_{26}$ | $a_{33}$ | $a_{43}$ | $a_{5N}$ | $a_{61}$ | $a_{73}$ | $a_{8N}$ | $a_{94}$ | $a_{101}$ |
| 10. | $a_{15}$ | $a_{26}$ | $a_{34}$ | $a_{43}$ | $a_{5N}$ | $a_{61}$ | $a_{7N}$ | $a_{8N}$ | $a_{91}$ | $a_{101}$ |
| 11. | $a_{16}$ | $a_{25}$ | $a_{31}$ | $a_{43}$ | $a_{54}$ | $a_{61}$ | $a_{7N}$ | $a_{81}$ | $a_{91}$ | $a_{102}$ |
| 12. | $a_{15}$ | $a_{26}$ | $a_{39}$ | $a_{42}$ | $a_{54}$ | $a_{61}$ | $a_{73}$ | $a_{81}$ | $a_{95}$ | $a_{105}$ |
| 13. | $a_{16}$ | $a_{26}$ | $a_{34}$ | $a_{43}$ | $a_{54}$ | $a_{61}$ | $a_{73}$ | $a_{81}$ | $a_{91}$ | $a_{101}$ |
| 14. | $a_{16}$ | $a_{25}$ | $a_{33}$ | $a_{42}$ | $a_{5N}$ | $a_{61}$ | $a_{7N}$ | $a_{81}$ | $a_{91}$ | $a_{105}$ |
| 15. | $a_{14}$ | $a_{26}$ | $a_{34}$ | $a_{42}$ | $a_{54}$ | $a_{61}$ | $a_{7N}$ | $a_{81}$ | $a_{91}$ | $a_{102}$ |
| 16. | $a_{14}$ | $a_{25}$ | $a_{34}$ | $a_{42}$ | $a_{54}$ | $a_{61}$ | $a_{7N}$ | $a_{8N}$ | $a_{94}$ | $a_{105}$ |
| 17. | $a_{16}$ | $a_{26}$ | $a_{31}$ | $a_{43}$ | $a_{54}$ | $a_{61}$ | $a_{7N}$ | $a_{8N}$ | $a_{95}$ | $a_{105}$ |
| 18. | $a_{14}$ | $a_{25}$ | $a_{39}$ | $a_{43}$ | $a_{5N}$ | $a_{61}$ | $a_{73}$ | $a_{81}$ | $a_{91}$ | $a_{101}$ |
| 19. | $a_{15}$ | $a_{25}$ | $a_{33}$ | $a_{42}$ | $a_{54}$ | $a_{61}$ | $a_{73}$ | $a_{8N}$ | $a_{91}$ | $a_{102}$ |
| 20. | $a_{18}$ | $a_{26}$ | $a_{34}$ | $a_{42}$ | $a_{5N}$ | $a_{61}$ | $a_{73}$ | $a_{8N}$ | $a_{91}$ | $a_{105}$ |
| 21. | $a_{16}$ | $a_{25}$ | $a_{33}$ | $a_{42}$ | $a_{5N}$ | $a_{61}$ | $a_{73}$ | $a_{8N}$ | $a_{95}$ | $a_{101}$ |
| 22. | $a_{16}$ | $a_{25}$ | $a_{34}$ | $a_{43}$ | $a_{54}$ | $a_{61}$ | $a_{73}$ | $a_{8N}$ | $a_{94}$ | $a_{101}$ |
| 23. | $a_{15}$ | $a_{25}$ | $a_{33}$ | $a_{42}$ | $a_{54}$ | $a_{61}$ | $a_{7N}$ | $a_{81}$ | $a_{94}$ | $a_{101}$ |
| 24. | $a_{18}$ | $a_{25}$ | $a_{38}$ | $a_{43}$ | $a_{54}$ | $a_{61}$ | $a_{73}$ | $a_{81}$ | $a_{94}$ | $a_{105}$ |
| 25. | $a_{15}$ | $a_{26}$ | $a_{38}$ | $a_{42}$ | $a_{54}$ | $a_{61}$ | $a_{7N}$ | $a_{8N}$ | $a_{91}$ | $a_{101}$ |
| 26. | $a_{18}$ | $a_{25}$ | $a_{34}$ | $a_{42}$ | $a_{5N}$ | $a_{61}$ | $a_{73}$ | $a_{81}$ | $a_{95}$ | $a_{102}$ |
| 27. | $a_{14}$ | $a_{25}$ | $a_{38}$ | $a_{43}$ | $a_{5N}$ | $a_{61}$ | $a_{7N}$ | $a_{8N}$ | $a_{95}$ | $a_{102}$ |
| 28. | $a_{15}$ | $a_{25}$ | $a_{31}$ | $a_{43}$ | $a_{5N}$ | $a_{61}$ | $a_{73}$ | $a_{8N}$ | $a_{91}$ | $a_{105}$ |
| 29. | $a_{16}$ | $a_{26}$ | $a_{39}$ | $a_{42}$ | $a_{5N}$ | $a_{61}$ | $a_{7N}$ | $a_{8N}$ | $a_{94}$ | $a_{102}$ |
| 30. | $a_{14}$ | $a_{26}$ | $a_{33}$ | $a_{43}$ | $a_{5N}$ | $a_{61}$ | $a_{7N}$ | $a_{81}$ | $a_{91}$ | $a_{105}$ |
| 31. | $a_{15}$ | $a_{26}$ | $a_{31}$ | $a_{43}$ | $a_{5N}$ | $a_{61}$ | $a_{73}$ | $a_{81}$ | $a_{94}$ | $a_{102}$ |
| 32. | $a_{15}$ | $a_{25}$ | $a_{34}$ | $a_{43}$ | $a_{5N}$ | $a_{61}$ | $a_{7N}$ | $a_{81}$ | $a_{95}$ | $a_{101}$ |

# Appendix C. Variation in Average Criteria

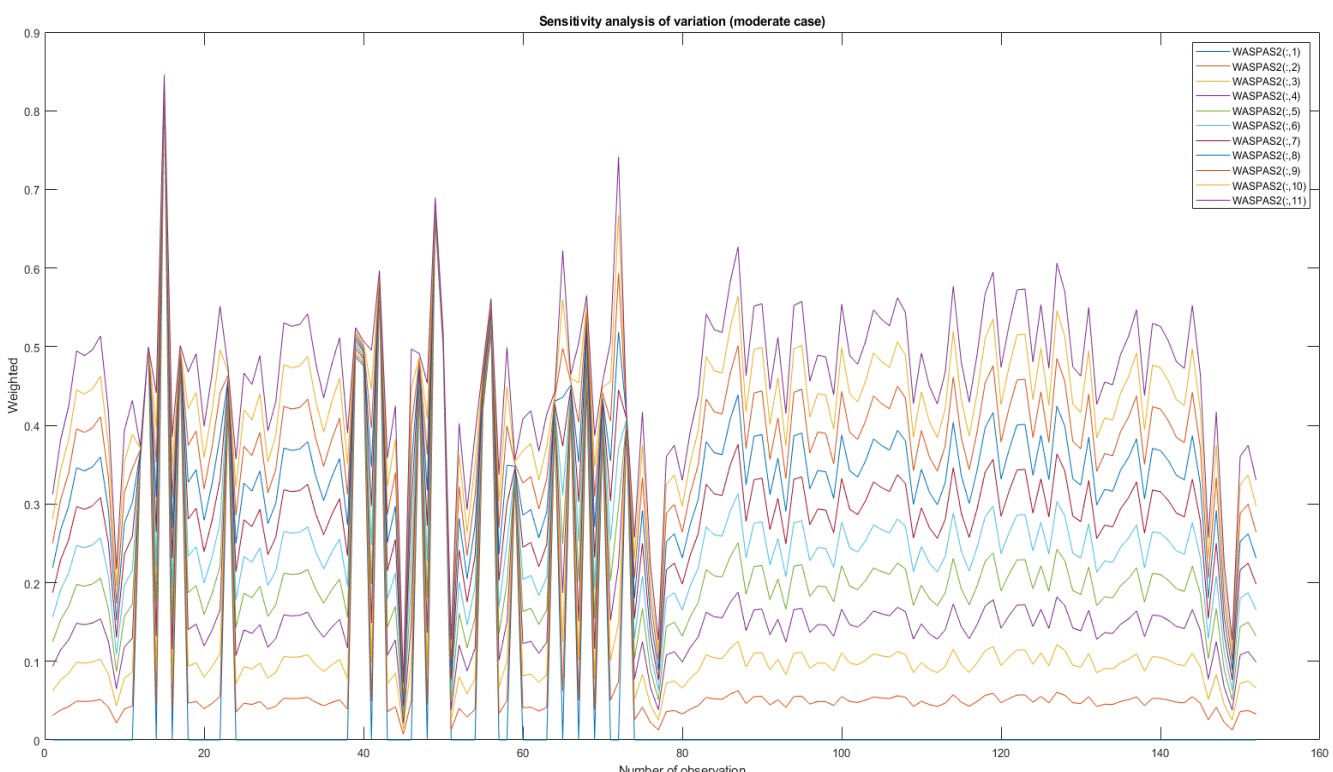

# Appendix D. Variation in Cost Criteria

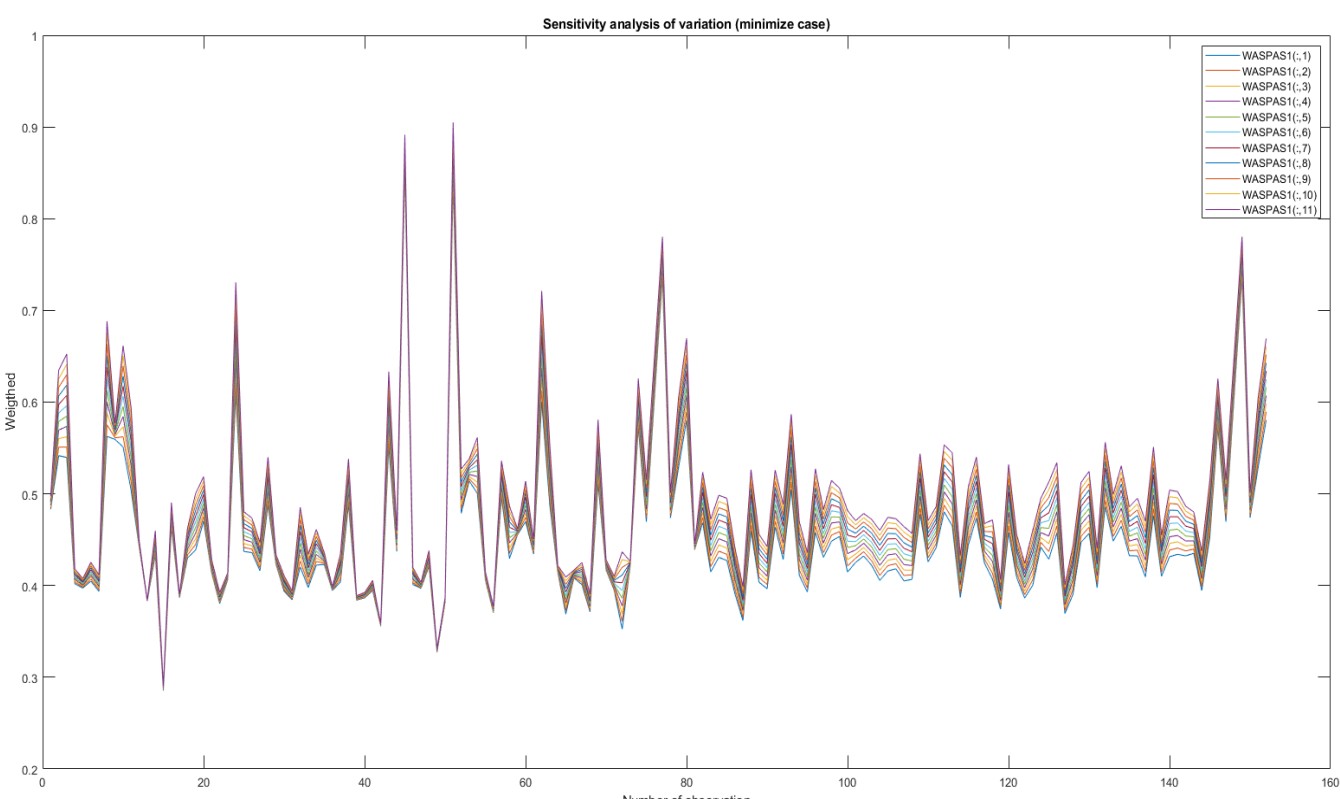

## Appendix E. Distribution of Average Criteria

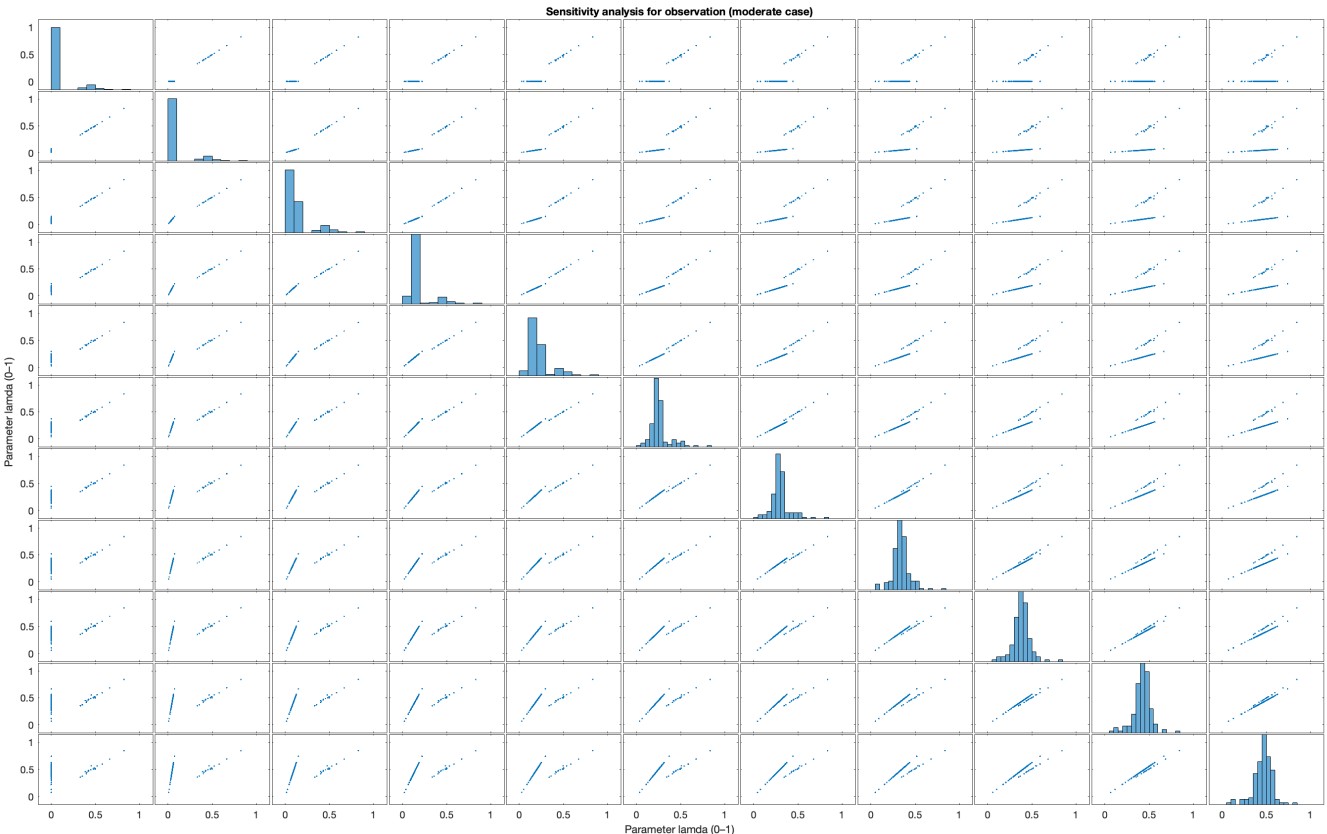

## Appendix F. Distribution of Cost Criteria

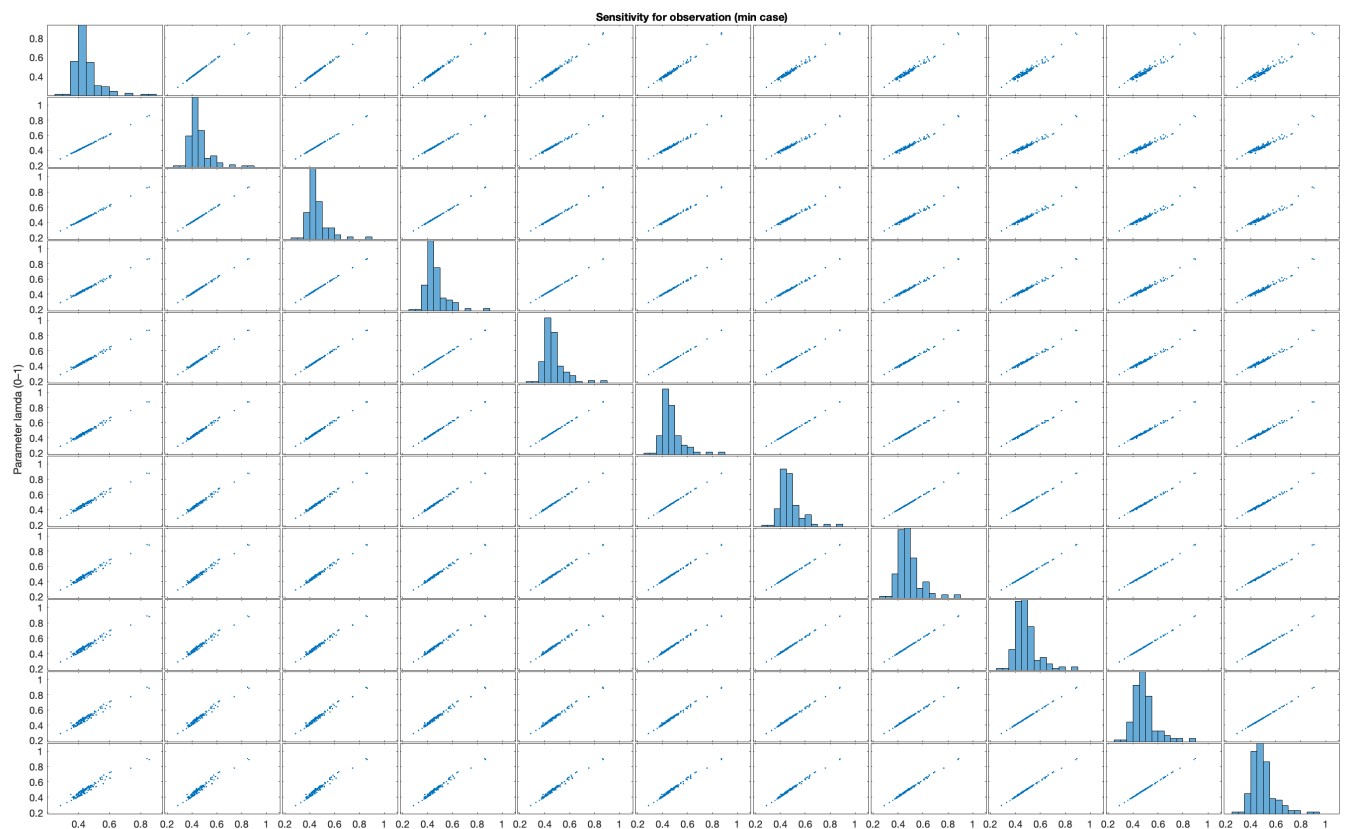

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
