# Peer review of "Threat Utility of the Seaport Risk Factors: Use of Rough Set-Based Genetic Algorithm"

_jmse, doi:10.3390/jmse10101484_

Round 1
Reviewer 1 Report
This paper quantitively assess the impacts of seaport risk factors (SRFs) on the supply chain disruptions (SCDs). It summarises 10 threats regarding SRFs and 61 SCD events based on literature. To quantify these impacts, utility functions of threats are expressed as the linear functions of SCD events. Based on utility functions, seaport risk threats can be ranked/prioritised. In the empirical study, a online questionnaire survey through face-to-face interviews was conducted, with a stratified random sampling targeting cargo owners, freight forwarders, ship owners, and ship management companies. The results of 150+ industry professionals were collected.
Overall, the research question is intersting, the methodology is sound, and the empirical findings are comprehensive and interesting.
The following comments could be helpful for improving the manuscript:
1. Introduction: move some discussion in Litature Review to enrich the background/motivation discussion.
2. Literature Review: at the end of the literature, research gaps and contributions to existing literature should be summarised. Apart from that, the latest relevant articles should be reviewed. For instance:
Nguyen, Son, Peggy Shu-Ling Chen, and Yuquan Du. "Risk assessment of maritime container shipping blockchain-integrated systems: An analysis of multi-event scenarios." Transportation Research Part E: Logistics and Transportation Review 163 (2022): 102764.
Nguyen, Son, Peggy Shu-Ling Chen, and Yuquan Du. "Container shipping operational risks: an overview of assessment and analysis." Maritime Policy & Management 49, no. 2 (2022): 279-299.
3. "Ship operators": I did not see the target group of "ship operators" in your survey, have "ship owner" and/or "ship management companies" already covered this? Note that in industry, these three parties could be different.
4. Shorten the technical/mathematical details in Section 3, because these discussions are general without touching too much the research question under investigation.
Author Response
First of things, thank you very much for your valuable comments on our manuscript. We have modified our manuscript according to the reviewers’ comments.
Reviewer #1:
This paper quantitively assess the impacts of seaport risk factors (SRFs) on the supply chain disruptions (SCDs). It summarizes 10 threats regarding SRFs and 61 SCD events based on literature. To quantify these impacts, utility functions of threats are expressed as the linear functions of SCD events. Based on utility functions, seaport risk threats can be ranked/prioritized. In the empirical study, an online questionnaire survey through face- to-face interviews was conducted, with a stratified random sampling targeting cargo owners, freight forwarders, ship owners, and ship management companies. The results of 150+ industry professionals were collected.
Overall, the research question is interesting, the methodology is sound, and the empirical findings are comprehensive and interesting.
The following comments could be helpful for improving the manuscript:
- Introduction: move some discussion in Literature Review to enrich the background/motivation discussion.
Thank you very much for the suggestion. We agree that some of discussions in the Literature Review are possibly to move in the Introduction. Hence, the moving paragraph is shown as follows:
…The several seaport risk factors and their resulting impact on supply chain continuity have increased the importance given to seaport operations. (Line 35 – 37)
The traditional seaport business is very labor-intensive and shares an interdependent relationship with other supply chain entities. The process by which this relationship is established poses a dimensional threat to the seaport supply chain, and hence plays a crucial role in seaport disruption events. This relationship process is associated with manpower in terms of soft skills and is affected by governmental policies. For in-stance, inefficient maritime security inspections at sea increase the shippers’ exposure to liabilities under the contract of carriage and lead to operational delays, which, in turn, lower the promptness of goods [4–5]. Similarly, miscommunication prevents the effective execution of instructions [6], and labor shortages delay cargo handling [5]. Moreover, a conflict of interest between parties hinders decision-making processes and disintegrates the supply chains [7]. Conventionally, a risk matrix is adopted to manage these supply chain risks. This risk matrix is classified into independent categories such as physical, financial, information, relationship, and innovation threats [8–10]. However, [11] argued that risk assessment/evaluation and risk treatment following the conventional risk identification techniques fail to account for complex dynamics across risks and risk sources, and hence yield sub-optimal solutions. Thus, we define the port-centric supply chain risk disruption (PSCRD) as the sum of the disruption level by the conditional seaport risk factor that influences the satisfaction of port-centric supply chain continuity. (Line 38 – 55)
- Literature Review: at the end of the literature, research gaps and contributions to existing literature should be summarized. Apart from that, the latest relevant articles should be reviewed.
Thank you very much for the comment. We adjusted the related research gap and emphasize our contribution to this matter. It is shown as follows:
The literature shows that seaports play a key role in supply chain continuity, given the increasing integration of seaports into supply chains. However, there is no explicit risk model to explain the interdependency between conditional seaport risk factors and potential threat of supply chain as well as to what extend the satisfaction level of seaport operation due to the causal connection. (Line 138 – 142)
Therefore, this study contributes to explain the causality connection among the many conditional seaport risk factors toward the potential threat of disrupted supply chain activities with a proposed framework to generate utility function. The utility function then useful to explain the satisfaction level of the relative importance of the conditional seaport risk factors. (Line 150 – 154)
Furthermore, the latest relevant articles were also added and shown in the line 82 – 91 and Table 1.
As an intersection between the worldwide mobility chain of goods and people, seaports have become critical to effectively and efficiently evaluate as well as manage PSCRD, protect the people and the environment, and maintain quality and performance. Recently, container shipping operations as the backbone of global supply chain creates a hotbed of multiple operational risks that affected to the others supply chain entities [14]. For example, [15] found that the low punctuality of delivery goods due to inefficient maritime security inspections increased exposure to liabilities under their contract of carriage. As an effect, stoppages for maritime security checks at sea generate delays, which raise shipping expenses such as reschedule services; including pilotage, class inspections, and planned maintenance. (Line 82 – 91)
- "Ship operators": I did not see the target group of "ship operators" in your survey, have "ship owner" and/or "ship management companies" already covered this? Note that in industry, these three parties could be different.
Thank you very much for the comment. We adjusted some explanation regarding the stakeholder categorization to make clear the stakeholder definition in our study.
The seaport-manager is a port authority whose works for a government agency or a government-owned company. The seaport-operator is in control of the seaport firm's operating procedure, which includes containers and non-containerized commodities such as automobiles, liquids, and dry bulk. Finally, seaport-users are stakeholders that collaborate with seaport operators and have a direct interest in the goods moved via seaports such as cargo owners, freight forwarders, ship owners, and ship management companies. (Line 429 – 437)
Regarding the “ship operator”, if the ship operator referred to the captain and the crew of the ship, both are included in “ship management companies”.
- Shorten the technical/mathematical details in Section 3, because these discussions are general without touching too much the research question under investigation.
The paper has two mayor goals in the methodology. Firstly, proposed the new framework in MCDM problem as well as explained the causality connection from the proposed method results. Secondly, explaining the interdependency phenomena between conditional seaport risk factors, potential threat of supply chain, and service level of seaport operation through the utility function. Hence, we shortened the sub-section 3.1 and deleted Fig. 1 that we think that it is common information in the rough set. (Line 170 – 212)
Reviewer 2 Report
Dear Authors,
thank you for the opportunity to read your article.
Please find below some comments for your consideration:
SSCRD - Seaport-centric supply chain risk disruption. Has the term port-centric been considered, and if yes, why are you using the term seaport-centric rather than port-centric? Several papers on port-centric logistics have been published and you have cited some of them in the literature review. A reasoning about the choice of the term and it's differences from port-centric should be provided.
Some background information about the context of Indonesian ports should be provided, as well as some reasoning about why this context, particularly since the geographical diversification of stakeholders is mentioned as a reason for further studies in the conclusion.
Line 15 &lines 280-285, 378-379 - Please specify the type of experts used.
Line 27 - please provide an example of such threats.
Lines 66-67: "Seaport business is very labor intensive and shares an interdependent relationship with other supply chain entities." --- Considering the development of automation in seaports and its adoption from several ports around the world, it might be more realistic to say: Seaport business traditionally has been very labor intensive...
Line 131 - please specify the source of the table as being authors' own
Line 392-393: "We employ an online questionnaire survey through face-to-face interviews." - The current wording is quite confusing and ambiguous as to what data collection method was used. Please clarify if these were real time interviews or if they were self-completed questionnaires.
Line 418 - Section 4.3 is called pilot test, however it describes reliability tests conducted after data collection. It should be clarified if the data collection instrument was piloted to a smaller sample prior the main data collection. Otherwise the title of the section should be changed to Reliability tests. Details about the pilot test should be provided prior to the analysis of the main dataset.
Line 724 - Galway and Dublin belong to the Republic of Ireland (aka Ireland) and not the United Kingdom
Line 725 - the doi is not correct as it results in a different paper.
Author Response
First of things, thank you very much for your valuable comments on our manuscript. We have modified our manuscript according to the reviewers’ comments.
Reviewer #2:
Thank you for the opportunity to read your article. Please find below some comments for your consideration:
- SSCRD - Seaport-centric supply chain risk disruption has the term port-centric been considered, and if yes, why are you using the term seaport-centric rather than port-centric? Several papers on port-centric logistics have been published and you have cited some of them in the literature review. A reasoning about the choice of the term and it's differences from port-centric should be provided.
Thank you very much for the valuable comment. Admittedly, our manuscript borrows the term of port-centric from several research articles. Then, we altered the term as port-centric supply chain risk disruption (PSCRD) to make the coherency from the previous research.
- Some background information about the context of Indonesian ports should be provided, as well as some reasoning about why this context, particularly since the geographical diversification of stakeholders is mentioned as a reason for further studies in the conclusion.
Thank you very much for the comment. We adjusted some explanation regarding the background information of the necessity of Indonesian context.
In the Indonesian context, an imbalance of cargo distribution, such as the availability of infrastructure, shipping patterns, supply and demand of maritime transport including port connectivity, between western part (developed economic region) and eastern part (developing economic region) make a challenge in the SSCRD. [15] addressed the issue of high logistics costs and price disparity between both regions. Moreover, this shipping cost harm the Gross Regional Domestic Product per capita in some part of developing economic due to the imparity above. (Line 123 – 129)
We select Indonesia as a case study due to its complexity that mentioned in the literature. [15] shows at least 9,755 cases of disruption management. Typical causes of disruption are "disobedience" in terms of operational rules, administrative regulations, and ministry decrees; weakness in the control systems, such as accounting and financial control; and policy. Both directly and indirectly, these factors relate to the export and import trade, as well as supply chain continuity and accidents with victims (either infrastructure or people). These phenomena reduce the seaport risk predictability. (Line 402 – 408)
- Line 15 &lines 280-285, 378-379 - Please specify the type of experts used.
Thank you very much for the correction. In order to make a clear, the explanation of experts used is adjusted as follows:
The seaport-manager is a port authority whose works for a government agency or a government-owned company. The seaport-operator is in control of the seaport firm's operating procedure, which includes containers and non-containerized commodities such as automobiles, liquids, and dry bulk. Finally, seaport-users are stakeholders that collaborate with seaport operators and have a direct interest in the goods moved via seaports such as cargo owners, freight forwarders, ship owners, and ship management companies. (Line 429 – 437)
- Line 27 - please provide an example of such threats.
The example of the threats is explained in the next paragraph as follows:
The traditional seaport business is very labor-intensive and shares an interdependent relationship with other supply chain entities. The process by which this relationship is established poses a dimensional threat to the seaport supply chain, and hence plays a crucial role in seaport disruption events. This relationship process is associated with manpower in terms of soft skills and is affected by governmental policies. For in-stance, inefficient maritime security inspections at sea increase the shippers’ exposure to liabilities under the contract of carriage and lead to operational delays, which, in turn, lower the promptness of goods [4–5]. Similarly, miscommunication prevents the effective execution of instructions [6], and labor shortages delay cargo handling [5]. Moreover, a conflict of interest between parties hinders decision-making processes and disintegrates the supply chains [7]. Conventionally, a risk matrix is adopted to manage these supply chain risks. This risk matrix is classified into independent categories such as physical, financial, information, relationship, and innovation threats [8–10]. However, [11] argued that risk assessment/evaluation and risk treatment following the conventional risk identification techniques fail to account for complex dynamics across risks and risk sources, and hence yield sub-optimal solutions. Thus, we define the port-centric supply chain risk disruption (PSCRD) as the sum of the disruption level by the conditional seaport risk factor that influences the satisfaction of port-centric supply chain continuity. (Line 38 – 55)
- Lines 66-67: "Seaport business is very labor intensive and shares an interdependent relationship with other supply chain entities." --- Considering the development of automation in seaports and its adoption from several ports around the world, it might be more realistic to say: Seaport business traditionally has been very labor intensive...
Thank you very much for the correction. We also agree with the reviewer suggestion and already altered as follows:
The traditional seaport business is very labor-intensive and shares an interdependent relationship with other supply chain entities… (Line 38-39)
- Line 131 - please specify the source of the table as being authors' own.
We adjusted it in line 157.
- Line 392-393: "We employ an online questionnaire survey through face-to-face interviews." - The current wording is quite confusing and ambiguous as to what data collection method was used. Please clarify if these were real time interviews or if they were self-completed questionnaires.
Thank you very much for the correction. The data collection is got from three procedures, such as:
… an online questionnaire survey, face-to-face interviews, and focused group discussion. (Line 423-424)
- Line 418 - Section 4.3 is called pilot test, however it describes reliability tests conducted after data collection. It should be clarified if the data collection instrument was piloted to a smaller sample prior the main data collection. Otherwise, the title of the section should be changed to Reliability tests. Details about the pilot test should be provided prior to the analysis of the main dataset.
Thank you very much for the valuable comment. We followed your suggestion. (Line 454)
- Line 724 - Galway and Dublin belong to the Republic of Ireland (aka Ireland) and not the United Kingdom
Thank you for the correction. We revised it. (Line 751)
- Line 725 - the doi is not correct as it results in a different paper.
We guess the reviewer pointed out the doi of reference [2] is not correct, but doi:10.21427/D7Q79J is the correct doi. However, we found the information of the authors to be wrong,, so we revised it. (Line 750)